# Antimicrobial Peptides for Skin Wound Healing

**DOI:** 10.3390/biom15111613

**Published:** 2025-11-17

**Authors:** Yifan Wu, Tingting Liu, Lili Jin, Chuyuan Wang, Dianbao Zhang

**Affiliations:** 1Department of Stem Cells and Regenerative Medicine, China Medical University, Shenyang 110122, China; 2022120047@cmu.edu.cn; 2Department of Pathology, The First Affiliated Hospital of Guangzhou Medical University, Guangzhou 510030, China; 3Department of Pharmacy, Liaoning Agricultural Technical College, Yingkou 115009, China; liutingting19881123@hotmail.com; 4School of Life Sciences, Liaoning University, Shenyang 110036, China; lilijin@lnu.edu.cn; 5Department of Endocrinology and Metabolism, The First Affiliated Hospital of China Medical University, Shenyang 110001, China

**Keywords:** antimicrobial peptides (AMPs), wound healing, keratinocytes, angiogenesis, collagen synthesis

## Abstract

Skin wound healing is a highly regulated biological process that requires the coordinated activity of multiple cell types. However, this process can be significantly impaired by factors such as metabolic diseases and infections, posing ongoing challenges for current treatment strategies. As a critical defense mechanism for cells and organisms against external threats, antimicrobial peptides (AMPs) hold great potential to enhance both the rate and quality of healing in both acute and chronic wounds. AMPs play a crucial role in promoting skin wound healing through mechanisms such as keratinocyte migration and proliferation, collagen synthesis and tissue remodeling, promotion of angiogenesis, immunomodulatory effects and broad-spectrum antimicrobial activity. Moreover, structural modifications and optimized delivery systems have further enhanced the stability and efficacy of AMPs. This paper explores the mechanisms by which AMPs aid in the healing of damaged skin and reviews the types of AMPs in clinical trials, providing a foundation for their development and clinical application.

## 1. Introduction

Serving as the largest organ of the human body, the skin operates as a sophisticated biomechanical and immunological interface that constitutes a primary physical barrier against environmental insults and, in parallel, orchestrates a constellation of homeostatic and innate-adaptive immune mechanisms to prevent pathogen infiltration, preserve systemic homeostasis, and preclude the onset of infection and disease [1,2]. When the skin’s natural barrier is compromised, wounds form, varying in severity and healing time [3,4]. Skin wound healing involves the coordinated actions of various cell types. This cascade is initiated by platelets, which aggregate to achieve hemostasis and secrete a milieu of growth factors that catalyze the healing response [5]. The ensuing inflammatory phase is characterized by the degranulation of mast cells, which release vasoactive amines such as histamine to promote vascular permeability and inflammation [6], and the recruitment of macrophages that phagocytose pathogens and debris while secreting pivotal cytokines and growth factors to direct subsequent repair stages [7]. During the proliferative phase, fibroblasts are activated to synthesize and deposit collagen and extracellular matrix components, constructing a provisional structural scaffold [8]. Concurrently, angiogenesis is driven by endothelial cells to form nascent vascular networks essential for perfusion and nutrient delivery to the nascent tissue [9]. The culmination of re-epithelialization is achieved through the proliferation and migration of keratinocytes, which restore the protective epidermal barrier [10]. This core cellular symphony is further supported by the contributions of adipocytes, immune cells (including T and B lymphocytes), stromal cells, and neural cells [11]. Collectively, these elements interact across the continuum of hemostasis, inflammation, proliferation, and remodeling to ensure efficacious wound closure and the restoration of tissue function.

Wounds are classified as either acute or chronic based on their cause and healing patterns [12,13]. Acute wounds are typically caused by sudden injuries, such as cuts, scrapes, or surgical incisions, and generally heal in a natural and orderly manner [14]. In contrast, chronic wounds, such as diabetic foot ulcers, venous leg ulcers, and pressure ulcers, often result from long-term health issues or sustained pressure, leading to prolonged inflammation and delayed healing, making treatment more challenging [15]. Current treatments for acute wounds primarily involve wound cleansing, dressing application, and the use of antibiotics when necessary [14], while chronic wound management requires necrotic tissue debridement, innovative dressings, and antimicrobial agents [16]. However, chronic wound treatment faces significant challenges, including biofilm formation, delayed healing, and antimicrobial resistance [17]. As antibiotic resistance increases, treating chronic wounds becomes even more difficult [15]. Therefore, there is an urgent need for innovative treatment strategies and emerging therapies to address these issues.

Antimicrobial peptides (AMPs), as an emerging therapeutic approach, exhibit broad-spectrum antimicrobial activity, modulate inflammatory responses, promote angiogenesis, and stimulate collagen deposition, effectively eliminating multidrug-resistant strains and accelerating wound healing. AMPs exert antimicrobial effects through various mechanisms, such as disrupting bacterial cell membrane integrity, inhibiting cell wall and nucleic acid synthesis, inducing oxidative stress, and interfering with cell division [18,19,20,21,22,23,24,25,26,27,28,29]. Beyond their direct antimicrobial action, AMPs also modulate the host immune response [30]. They enhance the secretion of cytokines and chemokines, which in turn promote phagocytosis and aid in the fight against infections [31]. It is noteworthy that AMPs serve as crucial components of the skin’s innate immune system, with magainin—first isolated from the skin of the African clawed frog *Xenopus laevis*—representing a classic example of such endogenous defense molecules [32]. AMPs activate immune cells, such as macrophages and T cells, facilitating the release of tumor necrosis factor-alpha (TNF-α) and interleukin-6 (IL-6), which attract additional immune cells to the wound site and improve phagocytic activity to hasten the elimination of pathogens [31]. Moreover, AMPs accelerate wound healing by promoting the migration and proliferation of keratinocytes through the activation of specific signaling pathways and cellular processes [33]. They trigger pathways such as the epidermal growth factor receptor (EGFR) and signal transducer and activator of transcription 3 (STAT3), which encourage cell proliferation and survival, as well as intracellular calcium mobilization and phospholipase C activation, thereby enhancing cellular functions [34]. In terms of angiogenesis, AMPs induce the expression of angiogenic factors like vascular endothelial growth factor (VEGF), which promotes endothelial cell proliferation and migration, aiding in the development of new capillary networks to ensure the wound receives adequate nutrients and oxygen [35]. Furthermore, AMPs stimulate fibroblast activity, increase collagen synthesis, and regulate matrix metalloproteinases (MMPs) activity, which promotes tissue remodeling and repair. This not only accelerates wound healing but also reduces scar formation [36]. As such, they are promising candidates for overcoming the complexities of chronic wound management and opening new avenues for future wound care [37,38].

AMPs are derived from animals, plants, microorganisms, and synthetic processes, with technological progress allowing their artificial production via chemical synthesis and genetic engineering. Notably, the incorporation of non-canonical amino acids has emerged as a powerful strategy to enhance their proteolytic stability and biological activity [39]. These innovations enhance their antimicrobial efficacy and stability [40,41,42]. Structural modifications, such as cyclization, fluorination, and D-amino acid substitution, further improve AMP stability and resistance to enzymatic degradation. Additionally, various delivery systems, including nanoparticles, liposomes, and hydrogels, are being developed to enhance AMP stability and targeted delivery in vivo, thereby optimizing their clinical efficacy and safety [43,44,45,46,47].

AMPs demonstrate significant potential in chronic wound management, particularly in treating infected diabetic foot ulcers, where they can notably shorten healing time and accelerate recovery. They also show promise in the treatment of skin conditions such as psoriasis, eczema, and skin cancers due to their abilities to modulate immune responses, exert anti-inflammatory effects, and promote tissue repair [48,49,50,51,52]. Ongoing research and technological progress are crucial for fully realizing the potential of AMPs in infection control, wound healing, and skin health improvement. Future studies will continue to explore the biochemical properties and functional roles of AMPs to enhance understanding of innate immunity and advance their clinical applications, including investigating molecular mechanisms, developing new AMPs and combination therapies, and optimizing delivery systems and formulations to improve therapeutic outcomes and safety. Despite current challenges, continuous research and technological progress are driving the development of safer and more effective clinical applications for AMPs, heralding potential breakthroughs in medical treatment and public health [53].

Management of burn wounds represents a significant challenge in wound care, characterized by extensive skin barrier defects, a high risk of secondary infection, and a hyperactive inflammatory response, often leading to delayed healing and pathological scar formation. Within this context, AMPs demonstrate a comprehensive therapeutic potential that extends beyond their fundamental antimicrobial functions. Research indicates that specific AMPs can directly intervene in key processes of burn healing. For instance, LL-37 has been shown to significantly promote the re-epithelialization of burn wounds by activating the epidermal growth factor receptor (EGFR) and its downstream ERK1/2 signaling pathway [54]. In terms of modulating the tissue repair microenvironment, peptides such as human β-defensin-3 (hBD-3) effectively induce angiogenesis, providing a crucial blood supply for the recovery of damaged tissue [55]. Of particular importance, certain AMPs (e.g., some frog-derived peptides and their synthetic analogs) exhibit the ability to regulate scar formation. Their mechanism involves the precise modulation of the transforming growth factor-beta (TGF-β) signaling pathway, specifically by inhibiting the pro-fibrotic TGF-β1 isoform [56,57]. This action reduces the excessive activation of fibroblasts and aberrant deposition of extracellular matrix, promotes more organized remodeling of collagen fibers, and ultimately improves the quality of healing [58]. Consequently, the application of AMPs in burn therapy represents a forward-looking strategy that integrates potent local immunity, direct acceleration of tissue repair, and improvement of long-term esthetic outcomes.

To provide a comprehensive and up-to-date foundation for this promising field, the literature presented in this review was identified through a systematic search of the PubMed, Web of Science, and Scopus databases, with a primary focus on literature from 2000 to 2025, while also including seminal works prior to this period. The search utilized key terms such as (“antimicrobial peptide”) AND (“wound healing”). This strategy, complemented by manual screening of reference lists, ensured the inclusion of over 290 pertinent studies that collectively illuminate the mechanisms, efficacy, and clinical potential of AMPs in wound healing, thereby solidifying the basis for the discussions herein (see Figure 1).

## 2. AMPs for Wound Healing

In the field of cutaneous wound healing, AMPs have emerged as a class of precision biotherapeutics with dual functionality, capable of concurrently addressing infection control and tissue regeneration [59]. Notably, their wound healing properties significantly extend beyond conventional antimicrobial roles [60]. Contemporary scientific investigation is systematically elucidating the molecular networks and cellular mechanisms underpinning this dual capacity [61]. Crucially, rigorous studies employing germ-free animal models have confirmed that AMPs possess intrinsic immunomodulatory functions independent of microbial clearance, including the precise regulation of immune cell chemotaxis, fine-tuning of cytokine networks, and direct activation of keratinocytes, fibroblasts, and endothelial cells [62,63].

Specifically, in genetically engineered germ-free murine models subjected to full-thickness skin injury, deficiency in the cathelicidin pathway (represented by murine CRAMP and its human ortholog LL-37) results in significantly impaired healing kinetics, characterized by delayed re-epithelialization and compromised neovascularization [64]. Critically, exogenous administration of LL-37 under strictly aseptic conditions completely rescues this healing deficit [64]. Concurrently, research has demonstrated that the platelet-derived antimicrobial chemokine PF-4/CXCL4 enhances angiogenic responses and promotes wound repair in completely sterile environments [65].

These findings collectively establish AMPs as multifunctional alarmins and pleiotropic regulators whose therapeutic potential is rooted in evolutionarily conserved immunomodulatory and cytotropic functions that remain fully operational even in the absence of microorganisms. Consequently, AMPs represent not merely potent antimicrobial agents but essential endogenous mediators orchestrating the complex pathophysiology of tissue repair. This deepened understanding provides a solid theoretical foundation for the rational design of a new generation of wound management strategies.

### 2.1. Broad-Spectrum Antimicrobial Activity

Broad-spectrum antimicrobial activity refers to the ability of a substance to effectively combat a wide range of pathogenic microorganisms, including bacteria, viruses, fungi, and parasites. AMPs are notable for their broad-spectrum activity due to their diverse mechanisms of action against different types of pathogens. This broad-spectrum activity is particularly valuable in combating multidrug-resistant strains and infections that are resistant to conventional antibiotics. The exploration of AMPs derived from different sources has highlighted their potential as versatile therapeutic agents in both preclinical and clinical settings (Table 1). These complex and synergistic multiple antimicrobial mechanisms, including membrane targeting attack, intracellular inhibition, anti-biofilm activity and immunomodulation, collectively form the foundation of their broad-spectrum activity (Figure 2).

#### 2.1.1. Human-Derived AMPs: Endogenous Defense and Repair Engines

Human β-defensin 3 (hBD-3) exhibits broad-spectrum antimicrobial activity in diabetic wound models, working through mechanisms such as activating Toll-like receptor signaling pathways, inhibiting glucocorticoid production, alleviating chronic inflammation, and stabilizing keratinocytes [66,67,68]. Salivary peptides promote wound healing and maintain microbial balance, with histatin 1 being particularly effective in acute skin wound healing [69,70]. LL-37 inhibits group A Streptococcus and shows promise for infection control in clinical trials, also exhibiting significant antimicrobial activity against *Pythium insidiosum* [71,72,73]. The DCD-1L peptide effectively inhibits *Acinetobacter baumannii* adhesion and biofilm formation, accelerating wound healing in mice [74]. Collagen VI-derived peptides (C6DP) demonstrate notable antibacterial activity against *Staphylococcus aureus*, *Escherichia coli*, and *Pseudomonas aeruginosa* [75]. Myxinidin2 and myxinidin3 promote wound healing in antibiotic-resistant infections by inhibiting inflammatory factors and regulating downstream mediators [76]. Laminin α3-derived LG4-5 peptides (Lα3-LG4-5 peptides) and Catestatin (Cst) play crucial roles in wound healing and antimicrobial defense, with Cst notably bridging neuroendocrine and cutaneous immune responses [77,78].

The endogenous human lactoferricin peptide, derived from the proteolytic fragment of the iron-binding protein lactoferrin, exhibits potent broad-spectrum antimicrobial activity [79]. It not only disrupts the outer membrane of Gram-negative bacteria but also binds to peptidoglycan and teichoic acids in Gram-positive bacteria, leading to increased membrane permeability and leakage of cellular contents. Its antimicrobial mechanisms further include iron deprivation, inhibition of bacterial adhesion, and endotoxin neutralization [79,80]. Similarly noteworthy is ubiquicidin, a class of cationic AMPs originating from the C-terminus of the ubiquitin protein. While functioning intracellularly in protein tagging, upon release into the extracellular environment, it demonstrates direct antimicrobial effects against bacteria (such as Staphylococcus aureus and *Klebsiella pneumoniae*) and fungi. Its mechanism of action involves interaction with bacterial membrane phospholipids and interference with intracellular processes [81].

#### 2.1.2. Naturally Sourced AMPs from Animals, Microbes and Plants

AMPs sourced from various animal origins have exhibited exceptional antibacterial and wound healing properties. For example, Scyreptin1-30, derived from *Scylla paramamosain*, demonstrates broad-spectrum antimicrobial activity and promotes wound healing in a murine burn infection model by disrupting bacterial membrane integrity and exhibiting strong anti-biofilm properties [82]. MPX, sourced from wasp venom, effectively kills bacteria by disrupting membrane integrity, increasing permeability, and altering membrane potential, significantly inhibiting *Staphylococcus aureus* colonization, reducing wound size and inflammation, and promoting healing in a mouse scratch model [83]. Cathelicidin-DM, from *Duttaphrynus melanostictus*, shows excellent bactericidal activity, killing bacteria within 15 min and demonstrating significant antimicrobial therapeutic potential in a mouse wound infection model [84]. Moreover, its antimicrobial action is independent of altering IL-6 and TNF-α secretion, indicating a direct mechanism of action [84,85].

Stigmurin, a linear peptide from the venom of *Tityus stigmurus*, exhibits significant antioxidant and antibacterial activities [86]. DMS-PS1 and DMS-PS2, derived from the waxy monkey tree frog, show broad-spectrum antimicrobial activity and effectively combat bacterial biofilms, significantly improving the healing of MRSA-infected wounds in mice [87]. The frog skin peptide Esculentin-1a(1-21)NH2(Esc(1-21)NH2) demonstrates broad-spectrum antimicrobial activity against the opportunistic Gram-negative bacterium *Pseudomonas aeruginosa*, with notable immunomodulatory properties [88]. Additionally, incorporating D-*Amino Acids* into Esc(1-21)NH2 reduces toxicity to mammalian cells, enhances effectiveness against biofilms, improves serum stability, and promotes lung epithelial cell migration [89]. Brevinin-2Ta exhibits significant antibacterial and anti-inflammatory effects against *Klebsiella pneumoniae*, promoting wound healing, reducing inflammation, and enhancing epithelial migration and angiogenesis [90]. Equine MSC secretome can inhibit the growth of *E. coli* and *S. aureus*, and through cysteine protease activity, it suppresses biofilm formation and disrupts mature MRSA biofilms, enhancing antibiotic efficacy in biofilm-related infections [91,92]. Magainins, isolated from the skin of the African clawed frog, inhibit the growth of bacteria and fungi [93].

PaTx-II, purified from the venom of the Australian King Brown Snake, exhibits moderate antimicrobial activity against Gram-positive bacteria (such as *S. aureus*, *E. aerogenes*, and *P. vulgaris*) by disrupting bacterial membrane integrity and forming pores. It also increases type I collagen levels in wound tissue and significantly reduces the levels of pro-inflammatory cytokines (such as IL-1β, IL-6, TNF-α, COX-2, and IL-10) [94]. Phospholipase A_2_ (svPLA_2_s) from Viperidae and Elapidae snake venoms have been proven effective against skin infections caused by *Staphylococcus aureus* (*S. aureus*). Topical application of svPLA_2_s can completely clear *S. aureus* within two weeks, demonstrating a dose-dependent bacteriostatic and bactericidal effect [95]. Snake venom peptides can bind to integrins with high affinity, inhibiting cell adhesion and accelerating wound healing. Additionally, by modulating the NF-κB signaling pathway, they influence the transforming growth factor (TGF)-β1/Smad pathway, reducing collagen production at wound sites, and thus present potential targets for drug development [96].

The marine AMP Epinecidin-1 (Epi-1) significantly reduces MRSA bacterial counts in the wounded area. Epi-1 not only lowers pro-inflammatory cytokine levels but also increases angiogenesis at the injury site, effectively combating MRSA infection [97,98]. *Tilapia piscidin* 3 (TP3) treatment doubled the survival rate of MRSA-infected mice, accelerated wound healing, and improved both antimicrobial and immunomodulatory responses [99].

Furthermore, AMPs derived from both plant and bacterial sources hold significant promise for therapeutic applications. Plantaricin A (PlnA), isolated from *Lactobacillus plantarum*, has been shown to synergistically improve the effectiveness of ciprofloxacin against multidrug-resistant *Staphylococcus aureus* and to foster wound healing in a skin wound infection model [100]. In a similar vein, PEGylated graphene oxide, when conjugated with OH-CATH30, has been found to expedite wound healing and diminish the presence of *Staphylococcus aureus* in a murine skin wound infection model, highlighting its potential as a therapeutic agent with excellent biocompatibility and drug-delivery properties [101]. The Ba49 peptide, isolated from *Bacillus subtilis* subsp. *spizizenii* found in onions, exhibits strong antibacterial activity against *Staphylococcus aureus* by altering membrane potential and inducing reactive oxygen species (ROS) production, with a prolonged post-antibiotic effect (PAE). At low concentrations, Ba49 peptide prevents biofilm formation, degrades mature biofilms, demonstrates intracellular killing potential against *S. aureus* in macrophages, and promotes fibroblast cell migration, indicating its wound healing efficacy [102]. Tyrothricin, an AMP from *Bacillus brevis*, has broad-spectrum activity and low resistance risk, primarily being applied in sore throat treatments and for minor infected wounds [103].

#### 2.1.3. Engineered and Synthetic AMPs: Rationally Designed Enhanced Weapons

Synthetic and engineered AMPs have exhibited unique advantages in antibacterial and wound healing research. For instance, the synthetic peptide DP1 exhibits no acute toxicity even at high doses and significantly promotes healing of *Staphylococcus aureus* infected wounds, reducing bacterial load and oxidative stress [104]. The engineered peptide Pep 6 effectively reduces bacterial load in *Escherichia coli* bacteremia and *Staphylococcus aureus* skin infection models and significantly inhibits MRSA biofilms [105]. Abhisin-like peptide (AB7) demonstrates broad-spectrum antimicrobial activity and effectively reduces bacterial load and pro-inflammatory mediators in a murine skin wound model [106]. The neuropeptide hHK-1 analog AH-4 shows strong antimicrobial activity and rapidly kills bacteria through membrane disruption [107]. The LEAP2 homolog Ll-LEAP2 is significantly upregulated following infection and exhibits selective antimicrobial activity against various bacteria by disrupting bacterial membranes and hydrolyzing bacterial gDNA [108]. The designed host defense peptide RP557 effectively inhibits bacterial growth in MRSA-infected wounds, reduces bacterial load, and accelerates wound healing in diabetic mice through topical application [109].

Tachyplesin I analog peptide (TP11A) effectively combats *Candida albicans*-*Staphylococcus aureus* poly-biofilm and mixed infections [110]. Peptides At3, At5, At8, and At10, derived from Ponericin-W1, exhibit high antimicrobial selectivity and activity, with At5 accelerating wound healing in a mouse infection model [111]. Recombinant peptides Trx-Ib-AMP4 and Trx-E50-52(rTrx) show synergistic antimicrobial effects against MRSA [112]. The chimeric peptide PP4-3.1 performs well against various bacteria and Candida, making it suitable for severe skin infections [113]. Gelatinase-responsive peptide GRAPN demonstrates strong photodynamic antimicrobial activity in a mouse model [114]. RV3 effectively kills *Pseudomonas aeruginosa* and inhibits inflammation [115]. Unnatural amino acid-based star-shaped poly(l-ornithine)s(UAASPLOs) exhibit broad-spectrum antimicrobial activity, efficiently disrupting biofilms and aiding in burn wound healing from *Pseudomonas aeruginosa* infection [116]. Novel CAMPs significantly reduce bacterial load in mouse skin wound models, with CAMP-A emerging as a promising alternative for treating *Pseudomonas aeruginosa* infections [117]. Collagenesis-Inducing Peptide 3.1-PP4(CIP 3.1-PP4) exhibits potent antibacterial and antibiofilm activity against multidrug-resistant Gram-negative bacteria, with low toxicity and collagenogenesis-inducing properties [118].

Pse-T2 effectively disrupts bacterial membranes and binds DNA, promoting faster healing of multidrug-resistant *Pseudomonas aeruginosa*-infected wounds and reducing inflammation more effectively than ciprofloxacin [119]. LI-F Type Peptide AMP-jsa9(LI-F AMP-jsa9) targets MRSA cell membranes, inhibits biofilm formation, reduces MRSA infections, and enhances wound healing by increasing VEGF and e-NOS expression while decreasing inflammatory cytokines [120]. SR-0379 is a functional peptide demonstrated to be safe, well-tolerated, and effective in clinical trials for chronic leg ulcers [121]. WRL3, an amphotericin peptide, inhibits MRSA biofilms and reduces bacterial burden in wound infections [122]. Chensinin-1b enhances antimicrobial activity by disrupting bacterial membranes and is effective in wound infection models [123]. Recombinant fusion protein BVN-Tβ4 promotes wound healing in diabetic mice [124], while PMO conjugates combined with thermoresponsive gels significantly improve healing of *Staphylococcus aureus*-infected wounds [125]. The synthetic decapeptide(SDP), when delivered using Pluronic F68 as a carrier, shows improved stability and activity, effectively reducing bacterial load [126]. Optimized end-tagged peptides derived from PRELP(OETP-PRELP) exhibit antimicrobial activity against various bacteria with low cell toxicity [127]. P-novispirin G10 demonstrates broad-spectrum antimicrobial activity and significantly reduces bacterial counts in a porcine skin wound model [128]. CPP-JDlys effectively suppresses MRSA proliferation [129], while AC7 shows significant anti-infective efficacy against drug-resistant *Pseudomonas aeruginosa* in a murine skin wound model, also reducing inflammation [130,131].

These studies emphasize the broad potential of AMPs in managing bacterial skin infections and facilitating wound healing. Continuous research and development are anticipated to further clarify the mechanisms of action and broaden the applications of these peptides, with the aim of offering innovative solutions for antibacterial therapy.

**Table 1 biomolecules-15-01613-t001:** Broad-Spectrum Antimicrobial Activity of AMPs.

Category	AMP Name	Source/Type	Key Antimicrobial Characteristics/Mechanisms	Demonstrated Efficacy in Wound Healing Models
Human-Derived	Human β-Defensin 3 (hBD-3)	Human	Broad-spectrum; activates TLR signaling; stabilizes keratinocytes [66,67,68]	Diabetic wound models [66]
	Salivary Histatin 1	Human Saliva	Promotes wound healing and maintains microbial balance [69,70]	Acute skin wound healing [70]
	LL-37 (hCAP-18)	Human (Cathelicidin)	Inhibits Group A Streptococcus and Pythium insidiosum; immunomodulatory [71,72,73]	Clinical trials for leg ulcers [71]; diabetic mouse models [73]
	DCD-1L	Human (Dermcidin)	Inhibits *Acinetobacter baumannii* adhesion and biofilm formation [74]	Mouse wound model [74]
	Collagen VI-derived Peptides (C6DP)	Human (Collagen VI)	Antibacterial activity against *S. aureus*, *E. coli*, and *P. aeruginosa* [75]	-
	Myxinidin2/Myxinidin3	Synthetic (Inspired by *N. flexuosus*)	Inhibits inflammatory factors; regulates downstream mediators [76]	Antibiotic-resistant infection models [76]
	Laminin α3-derived LG4-5 Peptides	Human (Laminin α3)	Crucial role in wound healing and antimicrobial defense [77]	-
	Catestatin (Cst)	Human (Chromogranin A)	Bridges neuroendocrine and cutaneous immune responses [78]	-
	Lactoferricin	Human (Lactoferrin)	Broad-spectrum; disrupts Gram−/− and Gram/+ membranes; iron deprivation [79,80]	-
	Ubiquicidin	Human (Ubiquitin)	Targets bacteria (e.g., *S. aureus*, *K. pneumoniae*) and fungi; interacts with membrane phospholipids [81,131]	-
Animal-Derived	Scyreptin1-30	Crab (*Scylla paramamosain*)	Broad-spectrum; disrupts membrane integrity; anti-biofilm [82]	Murine burn infection model [82]
	MPX	*Wasp Venom*	Disrupts membrane integrity and potential; kills *S. aureus* [83]	Mouse scratch model [83]
	Cathelicidin-DM	Toad (*Duttaphrynus melanostictus*)	Rapid bactericidal activity; mechanism independent of cytokine alteration [84,85]	Mouse wound infection model [84]
	Stigmurin	Scorpion (*Tityus stigmurus*)	Significant antioxidant and antibacterial activities [86]	-
	DMS-PS1/DMS-PS2	Frog (*Phyllomedusa distincta*)	Broad-spectrum; combats bacterial biofilms [87]	MRSA-infected wounds in mice [87]
	Esculentin-1a(1-21)NH2	Frog (*Pelophylax lessonae*)	Broad-spectrum against *P. aeruginosa*; immunomodulatory [88] D-amino acid version has enhanced stability [89]	-
	Brevinin-2Ta	Frog	Antibacterial and anti-inflammatory against *K. pneumoniae* [90]	Dermally wounded rats [90]
	Equine MSC Secretome	Horse (Mesenchymal Stem Cells)	Inhibits *E. coli* and *S. aureus*; suppresses/disrupts MRSA biofilms via cysteine protease [91,92]	-
	Magainins	Frog (*Xenopus laevis*)	Inhibits growth of bacteria and fungi [93]	-
	PaTx-II	Snake (*Pseudechis australis*)	Moderate activity vs. Gram−/+ bacteria; disrupts membranes; increases collagen I; anti-inflammatory [94]	-
	svPLA~2~s	Snake Venom (*Viperidae, Elapidae*)	Effective against *S. aureus* skin infections; dose-dependent bacteriostatic/bactericidal [95]	-
	Snake Venom Peptides	Snake Venom	Binds integrins; modulates NF-κB and TGF-β1/Smad pathways [96]	-
	Epinecidin-1 (Epi-1)	Fish (*Epinephelus coioides*)	Reduces MRSA counts; lowers pro-inflammatory cytokines; increases angiogenesis [97,98]	Mouse and swine burn wound models [97,98]
	*Tilapia Piscidin* 3 (TP3)	Fish (*Oreochromis niloticus*)	Improves survival, antimicrobial and immunomodulatory responses [99]	MRSA-infected mice [99]
Plant/Microbial-Derived	Plantaricin A (PlnA)	Bacterium (*Lactobacillus plantarum*)	Synergistic with ciprofloxacin against MDSA [100]	Skin wound infection model [100]
	OH-CATH30 (PEG-GO conjugate)	Fish (*O. niloticus*)/Synthetic Delivery	Graphene oxide conjugate enhances delivery; combats *S. aureus* [101]	Murine skin wound infection model [101]
	Ba49 Peptide	Bacterium (*Bacillus subtilis*)	Alters membrane potential; induces ROS; anti-biofilm; intracellular killing [102]	-
	Tyrothricin	Bacterium (*Bacillus brevis*)	Broad-spectrum; low resistance risk [103]	Minor infected wounds (clinical use) [103]
Engineered and Synthetic	DP1	Synthetic Peptide	No acute toxicity; reduces bacterial load and oxidative stress [104]	*S. aureus*-infected murine wounds [104]
	Pep 6	Engineered Peptide	Reduces bacterial load in bacteremia and skin infection; inhibits MRSA biofilms [105]	*E. coli bacteremia* and *S. aureus* skin infection models [105]
	Abhisin-like peptide (AB7)	Synthetic (Inspired by *A. halotolerans*)	Broad-spectrum; reduces bacterial load and pro-inflammatory mediators [106]	Murine skin wound model [106]
	AH-4 (hHK-1 analog)	Synthetic (*Neuropeptide Analogue*)	Strong activity; rapid membrane disruption [107]	-
	Ll-LEAP2	Fish (*Leptobrachium liui*)	Selective activity; disrupts membranes and hydrolyzes bacterial gDNA [108]	-
	RP557	Designed HDP	Inhibits bacterial growth in MRSA-infected wounds [109]	Diabetic mouse model (topical) [109]
	TP11A (Tachyplesin I analog)	Synthetic Analog	Combats C. albicans-*S. aureus* poly-biofilm and mixed infections [110]	-
	At5 (from Ponericin-W1)	Synthetic Derivative	High antimicrobial selectivity and activity [111]	Mouse infection model [111]
	Trx-Ib-AMP4/Trx-E50-52	Recombinant Peptides	Synergistic effects against MRSA [112]	-
	PP4-3.1 (Chimeric)	Synthetic Chimeric Peptide	Effective against various bacteria and Candida [113]	-
	GRAPN	Gelatinase-Responsive Peptide	Strong photodynamic antimicrobial activity [114]	Mouse model [114]
	RV3	Designed Peptide	Kills *P. aeruginosa* and inhibits inflammation [115]	-
	UAASPLOs	Unnatural Amino Acid Polymers	Broad-spectrum; disrupts biofilms [116]	*P. aeruginosa*-infected burn wounds [116]
	Novel CAMPs	Synthetic Cationic AMPs	Reduce bacterial load, CAMP-A vs. *P. aeruginosa* [117]	Mouse skin wound model [117]
	CIP 3.1-PP4	Collagenesis-Inducing Peptide	Potent vs. MDR Gram-negative bacteria; low toxicity [118]	-
	Pse-T2	Synthetic Peptide	Disrupts membranes and binds DNA; effective vs. MDR *P. aeruginosa* [119]	MDR *P. aeruginosa*-infected wounds [119]
	LI-F AMP-jsa9	Bacterium (*Paenibacillus polymyxa*)	Targets MRSA membranes; anti-biofilm; increases VEGF/e-NOS [120]	Murine scalded epidermis model [120]
	SR-0379	Functional Peptide	Safe, well-tolerated, and effective in clinical trials [121]	Chronic leg ulcers (clinical trial) [121]
	WRL3	Amphoteric Peptide	Inhibits MRSA biofilms; reduces bacterial burden [122]	Burn wound infection model [122]
	Chensinin-1b	Synthetic (Derived from *R. chensinensis*)	Disrupts bacterial membranes [123]	Wound infection model [123]
	BVN-Tβ4	Recombinant Fusion Protein	Promotes wound healing [124]	Diabetic mice [124]
	PMO Conjugates + Gel	Peptide-Morpholino Oligomer	Improves healing of *S. aureus*-infected wounds [125]	*S. aureus*-infected mouse wound model [125]
	Synthetic Decapeptide (SDP)	Synthetic Peptide	Improved stability and activity with Pluronic F68 carrier [126]	Restraint-stressed mice [126]
	OETP-PRELP	Optimized End-Tagged Peptides	Antimicrobial activity with low cell toxicity [127]	-
	P-novispirin G10	Recombinant Designer Peptide	Broad-spectrum; reduces bacterial counts [128]	Porcine skin wound model [128]
	CPP-JDlys	Cell-Penetrating Peptide Conjugate	Suppresses intracellular MRSA proliferation [129]	Keratinocytes infection model [129]
	AC7 (Abaecin Analog)	Rationally Designed	Significant efficacy against drug-resistant *P. aeruginosa* [130]	Murine skin wound model [130]

### 2.2. Immunomodulatory Effects

Immunomodulatory effects involve the modulation of the immune system’s activity, influencing how it responds to infections, injuries, and diseases. Peptides with immunomodulatory properties play a pivotal role in regulating immune responses by affecting various immune cells, signaling pathways, and cytokine production. These peptides can enhance or suppress immune functions, thereby influencing wound healing and inflammation. They contribute to the resolution of inflammation, promotion of tissue repair, and defense against pathogens by modulating cellular processes and immune signaling pathways. This broad range of immunomodulatory effects opens new avenues for therapeutic interventions in managing infectious and inflammatory conditions (Table 2).

LL-37 is induced in barrier organs during inflammation and infection and plays a crucial role in skin wound re-epithelialization but is absent in the epithelium of chronic ulcers [132]. Additionally, it promotes wound healing in diabetic mice by regulating TFEB-dependent autophagy, which is crucial for improving wound healing in high-glucose environments [133]. Human host defense peptides (HDPs) have broad antimicrobial and immunomodulatory properties, interacting with neutrophils, monocytes, and T cells to enhance cytokine production and neutralize lipopolysaccharides (LPS) [134]. The mast cell-specific receptor MRGPRX2/B2 is pivotal in host defense, as it activates mast cells to inhibit bacterial proliferation, prevent biofilm formation, mobilize neutrophils, and promote wound healing [135]. IDR-1018 promotes wound healing by modulating host immune pathways rather than through direct antibacterial activity, improving immune regulation in diabetic wounds [136].

Subsequently, the discussion encompasses peptides originating from amphibians. BugaCATH, derived from the toad *Bufo gargarizans*, accelerates wound healing by recruiting neutrophils and macrophages to initiate and expedite the inflammatory phase, regulating neutrophil phagocytosis, stimulating the production of cytokines and chemokines in macrophages, and promoting macrophage M2 polarization, which shifts the wound environment from a pro-inflammatory to an anti-inflammatory state. This process is crucial for reducing inflammation and effective wound healing, involving MAPK (ERK, JNK, p38) and NF-κB-NLRP3 signaling pathways [137]. Cathelicidin-OA1, another peptide sourced from amphibians, significantly accelerates skin wound re-epithelialization and granulation tissue formation by enhancing macrophage recruitment and inducing HaCaT cell proliferation and HSF cell migration. Although it lacks direct antimicrobial activity, its antioxidant properties play a crucial role in immune modulation [138].

Next, peptides derived from fish are introduced. *Salmo salar* skin collagen peptides (*Ss*-SCPs) and *Tilapia nilotica* skin collagen peptides (*Tn*-SCPs) promote wound healing by upregulating NOD2 and BD14 in wound tissue, reducing the expression of pro-inflammatory cytokines such as TNF-α, IL-6, and IL-8, and increasing the levels of anti-inflammatory cytokines like IL-10 [139]. Marine-derived AMPs, including Epinecidin-1 (Epi-1). Epi-1 reduces serum levels of pro-inflammatory cytokines TNF-α, IL-6, and MCP-1, and regulates the recruitment of monocytes and clearance of lymphocytes [97].

Additionally, the potential of peptides from various other sources is explored. Nisin A significantly reduces levels of pro-inflammatory cytokines (such as TNF-α, IL-6, and IL-8) and diminishes LPS-induced inflammatory responses, while also inhibiting MCP-1 production to decrease the recruitment of inflammatory cells, demonstrating strong immunomodulatory effects [140]. Andersonin-W1 (AW1) directly binds to Toll-like receptor 4 (TLR4), modulating the NF-κB signaling pathway to enhance inflammatory factor secretion, suppress LPS-induced excessive inflammation, and promote macrophage polarization, thus aiding re-epithelialization and angiogenesis [141]. TK-CATH, a novel anionic cathelicidin from the skin of *Tylototriton kweichowensis*, lacks direct antimicrobial activity but exhibits potent anti-inflammatory and wound healing properties, along with effective free radical scavenging and low cytotoxicity [142]. Esc(1-21)-1c (1 µM), similar to glucagon-like peptide-1, protects BRIN-BD11 cells from cytokine-induced apoptosis, enhances cell proliferation, stimulates insulin secretion, and reduces blood glucose levels, indicating its potential for managing type 2 diabetes [143]. Medicinal maggot excretions/secretions (ES) possess anti-inflammatory properties that aid corneal wound healing by reducing the production of inflammatory cytokines induced by Toll-like receptors [144].

Peptides originating from a wide array of sources exhibit significant potential in modulating immune responses, offering novel therapeutic avenues for the treatment of infectious and inflammatory conditions. Future research efforts should concentrate on further enhancing the therapeutic efficacy of these peptides and clarifying their mechanisms of action to aid in the development of new drugs.

**Table 2 biomolecules-15-01613-t002:** Main Effects of AMPs on Immunomodulation.

Category	AMP Name	Source	Key Immunomodulatory Mechanisms	Role in Wound Healing
Human-Derived	LL-37	Human (Cathelicidin)	Essential for re-epithelialization; absent in chronic ulcers [132]	Promotes healing in diabetic mice [133]
			Regulates TFEB-dependent autophagy in high-glucose environments [133]	
	Human Host Defense Peptides (HDPs)	Human	Interact with neutrophils, monocytes, and T cells to enhance cytokine production.	Broad defense and immune regulation.
			Neutralize LPS [134]	
	MRGPRX2/B2 Agonists	Human (Mast Cell Receptor)	Activates mast cells to inhibit bacterial proliferation and prevent biofilm formation.	Host defense and promotion of wound healing [135]
			Mobilizes neutrophils [135]	
	IDR-1018	Synthetic (Innate Defense Regulator)	Modulates host immune pathways rather than direct antibacterial activity.	Promotes healing in diabetic wounds [136]
			Improves immune regulation [136]	
Animal-Derived	BugaCATH	Toad (*Bufo gargarizans*)	Recruits neutrophils and macrophages.	Accelerates healing by initiating/expediting inflammation and resolving it [137]
			Regulates neutrophil phagocytosis.	
			Stimulates cytokine/chemokine production in macrophages and promotes M2 polarization via MAPK and NF-κB-NLRP3 pathways [137]	
	Cathelicidin-OA1	Frog (*Odorrana andersonii*)	Enhances macrophage recruitment.	Accelerates re-epithelialization and granulation tissue formation [138]
			Induces keratinocyte and fibroblast activity.	
			Possesses antioxidant properties [138]	
	Ss-SCPs/Tn-SCPs	Fish Skin Collagen	Upregulate NOD2 and BD14 in wound tissue.	Promote wound healing by modulating local inflammation and defense [139]
			Reduce pro-inflammatory cytokines (TNF-α, IL-6, IL-8).	
			Increase anti-inflammatory cytokine IL-10 [139]	
	Epinecidin-1 (Epi-1)	Fish (*Epinephelus coioides*)	Reduces serum levels of TNF-α, IL-6, and MCP-1.	Promotes healing in MRSA-infected burns [97,98]
			Regulates monocyte recruitment and lymphocyte clearance [97]	
	TK-CATH	Salamander (*Tylototriton kweichowensis*)	Potent anti-inflammatory and free radical scavenging activity.	Promotes wound healing with low cytotoxicity [142]
			Lacks direct antimicrobial activity [142]	
Other Sources	Nisin A	Bacterium (*Lactococcus lactis*)	Reduces pro-inflammatory cytokines (TNF-α, IL-6, IL-8).	Demonstrates strong immunomodulatory effects in wound healing [140]
			Diminishes LPS-induced inflammation.	
			Inhibits MCP-1 production [140]	
	Andersonin-W1 (AW1)	Insect (*O. andersonii*)	Binds directly to TLR4, modulating NF-κB pathway.	Aids re-epithelialization and angiogenesis in diabetic wounds [141]
			Suppresses LPS-induced excessive inflammation.	
			Promotes macrophage polarization [141]	
	Esc(1-21)-1c	Frog (Esculentin-1a derivative)	Protects pancreatic β-cells from cytokine-induced apoptosis.	Potential for managing type 2 diabetes, indirectly aiding diabetic wound healing [143]
			Enhances cell proliferation and insulin secretion [143]	
	Medicinal Maggot ES	Fly (*Lucilia sericata*)	Possesses anti-inflammatory properties.	Aids corneal wound healing [144]
			Reduces TLR-induced inflammatory cytokines [144]	

### 2.3. Collagen Synthesis and Tissue Remodeling

Dermal fibroblasts are pivotal in detecting pathogens via toll-like receptors, generating pro-inflammatory cytokines and chemokines, secreting growth factors and matrix metalloproteinases for tissue repair, and intensifying immune responses through communication with immune cells. Furthermore, they possess the ability to differentiate into adipocytes, offering a protective barrier against bacterial infections. A variety of peptides have been recognized for their capacity to boost wound healing by encouraging cell migration, the expression of growth factors, and the activation of fibroblasts. These peptides and signaling pathways underscore the diverse roles they play in facilitating effective wound healing and sustaining antimicrobial defense [145] (Table 3).

Human β-defensins (hBD-1, hBD-2, hBD-3, hBD-4) stimulate angiogenin secretion in dermal fibroblasts in a dose-dependent manner, mediated through EGFR, Src family kinases, JNK, p38, and NF-κB pathways [146]. Histatin 1 significantly enhances fibroblast migration and their transformation into myofibroblasts, activating the mTOR signaling pathway [147], while histatin 2 promotes fibroblast migration but has a minimal effect on proliferation [148]. LL-37 stimulates human dermal fibroblast migration in a time- and dose-dependent manner, increasing CXCR4 and SDF-1α expression [149].

Brevinin-2PN, a peptide originating from the dark-spotted frog (*Pelophylax nigromaculatus*), expedites the process of wound healing by encouraging the migration of human skin fibroblast cells and by boosting the expression of growth factor genes. Additionally, it exhibits antibacterial properties by compromising the integrity of bacterial cell membranes and by degrading the genomic DNA of pathogens [150]. PM-7, a peptide isolated from *Polypedates megacephalus*, aids in the healing of wounds in mice and promotes proliferation and migration in HUVEC and HSF cells via the MAPK signaling pathway [151]. Frog cathelicidin-NV fosters the proliferation of keratinocytes and fibroblasts, their differentiation into myofibroblasts, the production of collagen, and the secretion of essential cytokines involved in wound healing, all of which are mediated by the MAPK signaling pathways, without exerting direct antimicrobial effects or cytotoxicity [152].

The peptide Pt5-1c, derived from phosvitin, significantly speeds up the closure of wounds in fibroblast cultures in vitro, boosts dermal wound healing and re-epithelialization in mice in vivo, stimulates the migration and proliferation of fibroblasts, and facilitates collagen contraction by inducing the activation of fibroblasts into myofibroblasts [153]. Meanwhile, *Tilapia Piscidin* (TP)2-5 and TP2-6 significantly enhance the proliferation and migration of CCD-966SK fibroblasts while upregulating the expression of collagen I and III, thereby promoting tissue remodeling [154].

The AMP-IBP5 significantly induces fibroblast migration and proliferation by upregulating the LRP1 receptor [155,156]. *Aquaphilus dolomiae* extract (ADE-G2) enhances fibroblast proliferation, significantly promotes keratinocyte migration, and accelerates re-epithelialization of ex vivo skin explants [157]. Psoriasin (S100A7) and koebnerisin (S100A15) reduce extracellular matrix production and proliferation in human fibroblasts, with their expression significantly decreased in keloid tissue [158]. Synthetic human neutrophil peptide-1 (HNP-1) increases proalpha1(I) collagen mRNA and protein expression while decreasing MMP-1 levels, potentially aiding wound healing by enhancing extracellular matrix deposition and modulating its degradation [159].

In summary, these findings underscore the complex mechanisms that govern wound healing and antimicrobial defense. The peptides sourced from amphibians, humans, and fish, along with the pivotal roles played by vitamin D, the LRP1 receptor, and *Aquaphilus dolomiae* extract, collectively underscore the significance of signaling pathways and cellular interactions in fostering tissue repair and sustaining antimicrobial protection. These insights offer a more profound comprehension of the molecular underpinnings of wound healing and suggest potential therapeutic approaches for boosting tissue regeneration and combating infections.

**Table 3 biomolecules-15-01613-t003:** Main Effects of AMPs on Collagen Synthesis and Tissue Remodeling.

Category	AMP Name	Source	Key Effects on Fibroblasts and Tissue Remodeling	Demonstrated Efficacy in Wound Healing
Human-Derived	Human β-Defensins (hBD-1, -2, -3, -4)	Human	Stimulate angiogenin secretion in dermal fibroblasts via EGFR, Src, JNK, p38, and NF-kB pathways [146]	-
	Histatin 1	Human Saliva	Enhances fibroblast migration and transformation into myofibroblasts via mTOR signaling [147]	-
	Histatin 2	Human Saliva	Promotes fibroblast migration with minimal effect on proliferation [148]	-
	LL-37	Human (Cathelicidin)	Stimulates human dermal fibroblast migration in a time- and dose-dependent manner; increases CXCR4 and SDF-1α expression [149]	-
	AMP-IBP5	Human (IGFBP-5 derived)	Induces fibroblast migration and proliferation by upregulating the LRP1 receptor [155,156]	-
	Psoriasin (S100A7) and Koebnerisin (S100A15)	Human	Reduce extracellular matrix production and proliferation in human fibroblasts; expression decreased in keloid tissue [158]	Implicated in pathological scarring [158]
	Synthetic HNP-1	Human (Neutrophil Peptide-1)	Increases proalpha1(I) collagen mRNA and protein expression; decreases MMP-1 levels [159]	Potential aid in wound healing by enhancing ECM deposition [159]
Animal-Derived	Brevinin-2PN	Frog (*Pelophylax nigromaculatus*)	Encourages migration of human skin fibroblast cells; boosts expression of growth factor genes [150]	Expedites wound healing in models [150]
	PM-7	Frog (*Polypedates megacephalus*)	Promotes proliferation and migration in HSF cells via MAPK signaling pathway [151]	Aids healing of wounds in mice [151]
	Cathelicidin-NV	Frog (*Nanorana ventripunctata*)	Fosters proliferation of fibroblasts and their differentiation into myofibroblasts; promotes collagen production via MAPK pathways [152]	Accelerates wound healing in mice [152]
	Pt5-1c	Frog/Egg (*Phosvitin*-derived)	Stimulates migration and proliferation of fibroblasts; facilitates collagen contraction by activating fibroblasts into myofibroblasts [153]	Speeds up dermal wound healing and re-epithelialization in mice [153]
	*Tilapia Piscidin* (TP)2-5 and TP2-6	Fish (*Oreochromis niloticus*)	Enhance proliferation and migration of fibroblasts; upregulate expression of collagen I and III [154]	Promote tissue remodeling in wound models [154]
	*Aquaphilus dolomiae* extract (ADE-G2)	Bacterium	Enhances fibroblast proliferation and significantly promotes keratinocyte migration [157]	Accelerates re-epithelialization of ex vivo skin explants [157]

### 2.4. Promotion of Angiogenesis

Angiogenesis is the process by which new blood vessels are formed from pre-existing ones through various mechanisms, a crucial mechanism in wound healing and tissue repair that ensures an adequate supply of nutrients and oxygen to the injured area [9]. Numerous peptides have been thoroughly investigated for their capacity to foster wound healing and stimulate angiogenesis (Table 4). Among these, human β-defensins (hBDs), including hBD-1, hBD-2, hBD-3, and hBD-4, are notable for their antimicrobial properties and their ability to induce angiogenin secretion in dermal fibroblasts, with these effects mediated through EGFR, Src, JNK, p38, and NF-κB signaling pathways [146]. hBD-3 significantly enhances angiogenesis and increases the secretion of angiogenic growth factors, with its effects mediated through the FGFR1/JAK2/STAT3 signaling pathway [160]. Histatin-1 promotes wound healing by facilitating the migration of epithelial and endothelial cells [161]. Neurotensin, substance P, and insulin have significant effects on cell migration, with neurotensin and insulin increasing monocyte chemoattractant protein-1 levels and promoting angiogenesis [162].

Likewise, peptides of both animal origin and those synthesized in the laboratory hold considerable promise for advancing wound healing and stimulating angiogenesis. PM-7, a novel peptide derived from the *Polypedates megacephalus* frog, has shown its ability to promote wound healing in mice. This peptide fosters cell proliferation and migration in both HUVEC and HSF cells by activating the MAPK signaling pathway [151]. The *tilapia piscidin* (TP)2 peptides, particularly TP2-5 and TP2-6, have also exhibited significant pro-angiogenic properties, as demonstrated by their ability to enhance the migration of human umbilical vein endothelial cells (HUVEC) and promote neovascularization in vitro, indicating their strong potential for angiogenesis. In a murine model, topical application of TP2-5 and TP2-6 results in significantly reduced wound size by day 2 post-injury and accelerates wound healing compared to untreated wounds [154].

Nisin A significantly enhances the migration of human umbilical vein endothelial cells (HUVEC) and promotes neovascularization, while also indirectly affecting angiogenesis by reducing levels of pro-inflammatory cytokines such as tumor necrosis factor-α, interleukin-6, and interleukin-8 [140]. The novel angiogenic peptide AG30/5C, with antimicrobial properties, demonstrates good safety and efficacy in the treatment of severe leg ulcers [163]. The pro-angiogenic peptide proadrenomedullin N-terminal 20 peptide (PAMP) significantly promotes angiogenesis and re-epithelialization in both normoxic and ischemic wounds, and when used in combination with stem/progenitor cells, it restores wound contraction and effectively prevents widespread necrosis under ischemic conditions [164].

In summary, peptides derived from human and animal sources, as well as synthetic peptides, present a diverse array of mechanisms and benefits for angiogenesis. Each peptide possesses unique properties that contribute to the overall healing process, rendering them valuable candidates for therapeutic applications in the management of wounds.

**Table 4 biomolecules-15-01613-t004:** Main Effects of AMPs on Angiogenesis.

Category	AMP Name	Source	Pro-Angiogenic Mechanisms and Activities	Demonstrated Efficacy in Wound Healing
Human-Derived	Human β-Defensins (hBD-1, -2, -3, -4)	Human	Induce angiogenin secretion in dermal fibroblasts via EGFR, Src, JNK, p38, and NF-kB pathways [146]	-
	hBD-3	Human	Enhances angiogenesis and increases secretion of angiogenic growth factors via the FGFR1/JAK2/STAT3 pathway [160]	-
	Histatin-1	Human Saliva	Promotes wound healing by facilitating the migration of endothelial cells [161]	-
	Neurotensin, Substance P, Insulin	Human/Synthetic	Increase monocyte chemoattractant protein-1 levels and promote angiogenesis [162]	-
	Proadrenomedullin N-terminal 20 peptide (PAMP)	Human	Significantly promotes angiogenesis and re-epithelialization in both normoxic and ischemic wounds [164]	Restores wound contraction and prevents necrosis in ischemic conditions, especially with stem/progenitor cells [164]
Animal-Derived	PM-7	Frog (*Polypedates megacephalus*)	Fosters cell proliferation and migration in HUVECs via the MAPK signaling pathway [151]	Promotes wound healing in mice [151]
	*Tilapia Piscidin* (TP)2-5 and TP2-6	Fish (*Oreochromis niloticus*)	Enhance migration of HUVECs and promote neovascularization in vitro [154]	Topical application reduces wound size and accelerates healing in a murine model [154]
Other and Synthetic	Nisin A	Bacterium (*Lactococcus lactis*)	Enhances the migration of HUVECs and promotes neovascularization; indirectly affects angiogenesis by reducing pro-inflammatory cytokines [140]	-
	AG30/5C	Synthetic Angiogenic Peptide	Novel angiogenic peptide with inherent antimicrobial properties [163]	Demonstrated good safety and efficacy in a clinical study for severe leg ulcers [163]

### 2.5. Impact on Keratinocytes

Keratinocytes are the primary cells of the epidermis, constituting about 90% of the epidermal cell population [165,166]. They are crucial for epidermal formation through proliferation, differentiation, and migration, which are essential for re-epithelializing damaged skin [165,166]. Additionally, keratinocytes regulate inflammatory responses and tissue repair by secreting cytokines, chemokines, and growth factors, playing a key role in all stages of wound healing [167]. Their functions are regulated by multiple signaling pathways, including the epidermal growth factor receptor (EGFR), signal transducer and activator of transcription (STAT), and other critical molecular signaling pathways [160]. As a physical barrier and significant player in skin immune responses, keratinocytes are central to studies on wound healing [168,169,170].

During wound healing, AMPs enhance keratinocyte function and promote re-epithelialization through various signaling pathways and mechanisms (Table 5). Human beta-defensins (hBDs), particularly hBD-2, -3, and -4, activate EGFR and STAT3 signaling pathways, enhancing keratinocyte migration and proliferation, and stimulating the production of both pro-inflammatory and anti-inflammatory cytokines and chemokines [165,166]. hBD-2 facilitates wound healing by promoting EGFR and STAT3 phosphorylation, activating phospholipase C, and mobilizing intracellular Ca^2+^ [167], while hBD-3 accelerates wound healing through the FGFR/JAK2/STAT3 signaling pathway [160].

Histatins Hst1 and Hst2 promote keratinocyte migration and accelerate wound closure by activating the ERK1/2 pathway [168,169,170]. The AMP LL-37 protects keratinocytes from camptothecin (CAM)-induced apoptosis by reducing caspase-3 activity and upregulating the expression of cyclooxygenase-2 (COX-2) and inhibitor of apoptosis protein-2 (IAP-2), while also promoting migration through EGFR pathway activation [171,172]. Growth factors like IGF-I and TGF-α increase the expression of TLR5 and TLR9, and their activation enhances the production of IL-8 and human beta-defensins, thereby promoting keratinocyte migration and improving blood perfusion in a hindlimb ischemia model [54,173,174]. S100A7 exhibits a biphasic response to bacterial exposure, regulated by NFκB/p38MAPK, caspase-1, and IL-1α, with initial activation mediated by TLR signaling and chronic response dependent on caspase-8 downregulation [175]. AG-30/5C promotes keratinocyte migration and proliferation and enhances cytokine/chemokine production through MrgX receptors and the MAPK and NF-κB pathways [176]. SPINK9 enhances keratinocyte migration by inducing EGFR transactivation [177].

Esculentin-1a(1-21)NH2(Esc(1-21)NH2), derived from frog skin, significantly promotes keratinocyte migration through the activation of the EGF receptor and STAT3 protein, showing greater efficacy than LL-37, and supports wound healing without cytotoxicity to mammalian cells [178]. Cathelicidin-NV from the plateau frog *Nanorana ventripunctata* enhances keratinocyte and fibroblast proliferation, promotes collagen production, and accelerates re-epithelialization through the MAPK signaling pathway, without direct antimicrobial activity or cytotoxicity [152]. Cathelicidin-NV also protects HaCaT cells from UVB-induced photoaging by scavenging excessive intracellular ROS [179]. Temporins A and B promote keratinocyte migration and wound healing via the EGFR signaling pathway, and possess dual functions in antibacterial action and immune modulation [180]. AH90, derived from frog skin, accelerates wound healing in mice through the NF-κB and JNK MAPK pathways [181]. HB-107 treatment induces keratinocyte hyperplasia and leukocyte infiltration, accelerating wound healing in mice [182].

*Tilapia piscidin* (TP) 2-5 and TP 2-6 possess dual functionalities: they markedly stimulate cell proliferation and migration by activating EGFR signaling, and they also enhance the expression of collagen and growth factors, demonstrating angiogenic characteristics [154]. The gecko cathelicidin-related antioxidant peptide (Gj-CATH3 derivative) displays significant antioxidant activity and promotes wound healing, facilitating cell proliferation and diminishing oxidative stress [183].

A novel chimeric peptide with cell-penetrating capabilities, Tylotoin-sC18*, has been found to accelerate wound healing by boosting keratinocyte migration and proliferation, while also displaying antimicrobial properties [184]. Nisin A significantly enhances the migration of HaCaT keratinocytes without affecting proliferation, potentially by reducing levels of pro-inflammatory cytokines such as tumor necrosis factor-α, interleukin-6, and interleukin-8. Additionally, Nisin A treatment decreases monocyte chemoattractant protein-1 levels in HaCaT cells and increases re-epithelialization of porcine skin [140]. AMP-IBP5 significantly promotes keratinocyte migration and proliferation via Mas-related gene X receptors (MrgX1-X4) and upregulation of the LRP1 receptor, with its effects mediated through the MAPK and NF-κB pathways [155,156]. Additionally, Pep19-2.5 stimulates keratinocyte migration and ERK1/2 phosphorylation by activating the P2X7 receptor, which increases cytosolic calcium levels and mitochondrial ROS [185].

In conclusion, the integration of AMPs into wound healing protocols offers significant promise for boosting keratinocyte function, accelerating re-epithelialization, and regulating inflammatory responses. These peptides play a pivotal role in innate immunity and wound healing by effectively inducing cell migration, proliferation, and cytokine production via diverse signaling pathways. Additionally, the therapeutic potential of peptides opens up exciting possibilities for the treatment of chronic wounds and various dermatological disorders. To maximize their clinical effectiveness and enhance patient outcomes in cases of compromised wound healing, further investigation into their precise mechanisms of action and the development of optimal delivery systems is imperative (Figure 3).

**Table 5 biomolecules-15-01613-t005:** Main Effects of AMPs on Keratinocytes.

Category	AMP Name	Source	Key Effects on Keratinocytes and Mechanisms	Role in Wound Healing
Human-Derived	Human β-Defensins (hBD-2, -3, -4)	Human	Activate EGFR and STAT3 signaling, enhancing migration, proliferation, and cytokine production [165,166]	Facilitate re-epithelialization.
	hBD-2	Human	Promotes migration via EGFR/STAT3 phosphorylation, PLC activation, and intracellular Ca^2+^ mobilization [167]	Enhances wound closure.
	hBD-3	Human	Accelerates healing through the FGFR/JAK2/STAT3 signaling pathway [160]	Promotes keratinocyte migration and proliferation.
	Histatins (Hst1, Hst2)	Human Saliva	Promote migration and accelerate wound closure by activating the ERK1/2 pathway [168,169,170]	Critical for oral and skin wound closure.
	LL-37	Human (Cathelicidin)	Protects from apoptosis (reduces caspase-3, upregulates COX-2/IAP-2); promotes migration via EGFR transactivation [171,172]	Promotes re-epithelialization; deficient in chronic ulcers [132]
	S100A7 (Psoriasin)	Human	Exhibits biphasic response to bacteria via NF-κB/p38MAPK, caspase-1, and IL-1α; sustained secretion requires caspase-8 downregulation [175]	Involved in skin defense and homeostasis.
	SPINK9	Human	Enhances migration by inducing EGFR transactivation [177]	Contributes to epidermal repair.
	AMP-IBP5	Human (IGFBP-5 derived)	Promotes migration and proliferation via MrgX receptors and LRP1 upregulation, mediated by MAPK and NF-κB pathways [155,156]	Enhances re-epithelialization.
	Pep19-2.5	Synthetic	Stimulates migration and ERK1/2 phosphorylation by activating P2X7 receptor, increasing cytosolic Ca^2+^ and mitochondrial ROS [185]	Promotes keratinocyte migration.
Animal-Derived	Esculentin-1a(1-21)NH2	Frog (*Pelophylax lessonae*)	Promotes HaCaT migration via EGF receptor and STAT3 activation; greater efficacy than LL-37; no cytotoxicity [178]	Supports wound healing.
	Cathelicidin-NV	Frog (*Nanorana ventripunctata*)	Enhances keratinocyte proliferation and accelerates re-epithelialization via MAPK pathway [152] Also protects from UVB-induced photoaging [179]	Promotes cutaneous wound healing in mice [152]
	Temporins A and B	Frog (*Rana temporaria*)	Promote HaCaT migration and wound healing via the EGFR signaling pathway [180]	Dual antibacterial and immune modulatory functions.
	AH90	Frog Skin	Accelerates wound healing in mice through NF-κB and JNK MAPK pathways [181]	Potential wound healing-promoting peptide.
	*Tilapia Piscidin* (TP)2-5 and TP2-6	Fish (*Oreochromis niloticus*)	Stimulate cell proliferation and migration by activating EGFR signaling [154]	Contribute to wound healing with angiogenic properties.
	Gj-CATH3 Derivative	Gecko (*Gekko japonicus*)	Displays significant antioxidant activity and promotes wound healing, facilitating cell proliferation and diminishing oxidative stress [183]	Facilitates wound repair.
Other and Synthetic	AG-30/5C	Synthetic Angiogenic Peptide	Promotes migration, proliferation, and enhances cytokine/chemokine production through MrgX receptors and MAPK/NF-κB pathways [176]	Multifunctional role in healing.
	Tylotoin-sC18	Synthetic Chimeric Peptide	Accelerates healing by boosting keratinocyte migration and proliferation, with additional antimicrobial properties [184]	Chimeric peptide with cell-penetrating capability.
	Nisin A	Bacterium (*Lactococcus lactis*)	Significantly enhances migration of HaCaT keratinocytes without affecting proliferation; reduces pro-inflammatory cytokines and increases re-epithelialization [140]	Promotes healing in porcine skin models.

## 3. Clinical Trials

Table 6 offers a comprehensive summary of clinical trials involving various peptides and their mechanisms of action in accelerating wound healing. These peptides exhibit significant potential for the treatment of infectious wounds and ulcers by employing a range of mechanisms, such as disrupting bacterial membranes and modulating the immune response. For example, Gramicidin, which has been used in the treatment of infected wounds and ulcers, has propelled to Phase III clinical trials, where its efficacy is being assessed based on its membrane-disrupting and immune-modulating properties [186]. Polymyxin B, which targets Gram-negative bacteria, is also in Phase III clinical trials, utilizing mechanisms similar to those of membrane disruption and immune modulation [187]. Likewise, Daptomycin, indicated for the treatment of skin infections and bacteremia, is progressing through Phase III clinical trials, with its mode of action involving comparable mechanisms [188].

The human AMP LL-37, when applied to leg ulcers, exhibits its effects by disrupting cell membranes and modulating the immune response, and is currently undergoing Phase II clinical trials [189]. Melittin, the main constituent of bee venom, which is used to treat inflammation, also employs these mechanisms and is now in Phase I/II clinical trials [190]. Pexiganan (MSI-78), a derivative of Magainin, is specifically designed for the treatment of diabetic foot ulcers and is advancing through Phase III clinical trials, leveraging mechanisms of membrane disruption and immune modulation [191].

Furthermore, the synthetic peptide p2TA (AB103), currently in Phase III clinical trials, is being investigated for its potential in treating necrotizing tissue infections by modulating immune responses [192]. D2A21, which employs membrane disruption to combat infections in burn wounds, is also in Phase III clinical trials [193]. GSK1322322, a compound that inhibits peptidyl transferase enzymes to target bacterial skin infections, is undergoing Phase II clinical trials [194]. PMX-30063, an analog of defensin, is addressing acute bacterial skin infections by disrupting membranes and modulating the immune system, and is advancing through Phase II clinical trials [195]. XF-73, a peptide derivative of porphyrin, is being utilized for the treatment of Staphylococcus infections through membrane disruption and is currently in Phase II clinical trials [196]. Nisin, which targets Gram-positive bacteria by causing cell membrane depolarization, is in the early stages of clinical trials [197]. Lastly, PL-5, a synthetic peptide designed for skin infections, functions through membrane disruption and is currently in Phase I clinical trials [198].

Brilacidin is a structurally optimized small-molecule antimicrobial peptide mimetic inspired by host defense peptides, particularly human defensins [199]. It mimics the cationic amphipathic structure of natural AMPs and disrupts bacterial membrane integrity via a non-lytic mechanism, thereby exerting rapid bactericidal effects [199]. Additionally, it possesses anti-inflammatory and epithelial repair-promoting functions. In clinical studies, Brilacidin has advanced to Phase II and III trials for the treatment of acute bacterial skin and skin structure infections (ABSSSI) and oral mucositis [200]. Its advantages include broad-spectrum antimicrobial activity (including efficacy against drug-resistant strains such as MRSA), a low propensity to induce bacterial resistance, and favorable local tolerability and systemic safety profile, highlighting its significant potential as a next-generation topical anti-infective therapeutic agent [200].

These peptides exhibit significant therapeutic promise in clinical contexts, elucidating their diverse mechanisms in enhancing wound healing and antimicrobial defense, thereby offering crucial insights for future research.

**Table 6 biomolecules-15-01613-t006:** Current Status and Developments in Clinical Trials of AMPs.

Name	Target	Administration	Phase	ID	Mechanism
Gramicidin [186]	Infected wounds and ulcers	Topical	III	NCT00534391	Membrane disruption/immunomodulation
Polymyxin B [187]	Gram-negative bacteria	Topical	III	NCT00490477; NCT00534391	Membrane disruption/immunomodulation
Daptomycin [188]	Skin infection/bacteremia	Intravenous	III	NCT01922011; NCT00093067; NCT01104662; NCT02972983	Membrane disruption/immunomodulation
LL-37 [189]	Leg ulcers	Topical	II	EUCTR2012-002100-41	Membrane disruption/immunomodulation
Melittin [190]	Inflammation	Intradermal	I/II	NCT02364349, NCT01526031	Membrane disruption/immunomodulation
Pexiganan (MSI-78) [191]	Diabetic foot ulcers	Topical	III	NCT00563394; NCT00563433; NCT01590758; NCT01594762	Membrane disruption/immunomodulation
p2TA (AB103) [192]	Necrotic tissue infection	Intravenous	III		Immunomodulation
D2A21 [193]	Burn wound infections	Topical	III		Membrane disruption
GSK1322322 [194]	Bacterial skin infection	Oral	II	NCT01209078	Peptide deformylase inhibitor
PMX-30063 [195]	Acute bacterial skin infection	Intravenous	II	NCT01211470; NCT02052388	Membrane disruption/immunomodulation
XF-73 [196]	Staphylococcal infection	Topical	II	NCT03915470	Membrane disruption
Nisin [197]	Gram-positive bacteria	Oral		NCT02928042; NCT02467972	Depolarization of cell membrane
PL-5 [198]	Skin infections	Topical	I		Membrane disruption
Brilacidin [199,200]	ABSSSI	Intravenous/Oral	II/III	NCT02052388; NCT04240223; NCT04784897	Membrane disruption/PDE4 inhibitor

## 4. Peptide Formulations

Over the past few years, numerous sophisticated biomaterials, including hydrogels, have been utilized in wound care to enhance treatment efficacy at various healing stages [201]. Concurrently, nanosystems, through mechanisms such as encapsulation, targeting, and sustained release, improve the delivery, stability, and efficacy of AMPs for both systemic and topical applications [43]. Specific instances have shown encouraging results in preclinical and clinical environments, offering a supportive matrix that can bolster the delivery and efficacy of AMPs in facilitating wound healing.

### 4.1. Hydrogels

Recent breakthroughs in hydrogel-based therapies that incorporate AMPs have shown considerable promise in the field of wound care. These AMP hydrogels can be classified according to their functional properties and applications, with each type uniquely contributing to the healing process.

Self-healing and multifunctional hydrogels have exhibited significant potential in biomedical applications. For example, a sponge patch based on peptide hydrogels, which incorporates AMPs and medical agarose, displays outstanding antibacterial properties and fosters wound healing in both in vitro and animal models [202]. Furthermore, the DA7CG@C macroporous hydrogel, which combines the AMP DP7 with placenta-derived mesenchymal stem cells (PMSCs), markedly enhances wound healing and extracellular matrix remodeling [203]. Another multifunctional hydrogel, consisting of gelatin, chitosan, histatin-1, and polypyrrole-based nanoparticles, hastens diabetic wound healing by encouraging angiogenesis and anti-inflammatory responses [204]. These self-healing and multifunctional hydrogels not only expedite wound healing but also offer antimicrobial protection, highlighting their promise in the management of complex wounds.

Injectable hydrogels provide versatile application methods and have shown effective treatment outcomes in a variety of wound conditions [201]. For instance, near-infrared light-responsive injectable hydrogels, which incorporate dopamine and PEG-functionalized gellan gum, exhibit synergistic antibacterial effects against pathogens such as *Pseudomonas aeruginosa* and *Staphylococcus aureus*, indicating their potential in treating infected wounds [205]. Furthermore, the PNI/RA-Amps/E hydrogel, a combination of RADA16-Amps and MGF E peptide, enhances wound healing through its antibacterial properties and sustained peptide release, suggesting its promise as an innovative wound dressing [206]. Another example comprises AMP-based hydrogel patches that exhibit significant antimicrobial activity and promote wound healing [202]. These injectable hydrogels not only offer adaptable application methods but also provide effective treatment for a range of wound conditions.

pH-sensitive and responsive hydrogels are pivotal in managing drug release and optimizing conditions for wound healing. For example, a pH-sensitive mixed hydrogel that integrates oxidized β-D-glucan with the AMP C8G2 exhibits significant antibacterial activity and excellent biocompatibility [207]. Another pH-responsive peptide hydrogel, FHHF-11, modulates its firmness and boosts antimicrobial capabilities, rendering it appropriate for a range of wound care applications [208]. Moreover, pH-switchable antibacterial hydrogels, which are based on self-assembling pentapeptides, effectively eliminate biofilms and foster wound healing by stimulating angiogenesis and collagen synthesis [209]. The DP7-ODEX pH-sensitive hydrogel, which merges AMP DP7 with oxidized dextran, effectively combats multidrug-resistant bacteria and encourages scar-free wound healing [210]. These hydrogels adapt their functionalities in response to pH changes, thereby enhancing therapeutic efficacy.

Antibacterial and anti-infection hydrogels are highly effective in managing wound infections. For instance, hydrogels containing the AMP NZ2114, which are based on hydroxypropyl cellulose (HPC) and sodium alginate (SA), showcase sustained release and significant antimicrobial efficacy, rendering them ideal for treating infected wounds [211]. Of particular significance in antimicrobial hydrogel design is the integration of silver-based antimicrobial systems, which encompass both silver nanoparticles (AgNPs) and controlled-release silver ions (Ag^+^) [212]. These systems operate through distinct yet complementary mechanisms: AgNPs exert antibacterial activity via direct contact-mediated membrane disruption, sustained release of bioactive Ag^+^, and generation of reactive oxygen species (ROS), whereas silver ions (Ag^+^) directly impair bacterial respiratory chain function, compromise structural membrane integrity, and inactivate essential enzymatic systems, collectively contributing to their rapid and broad-spectrum antimicrobial profile [212]. The strategic combination of these functional silver species with AMPs within hydrogel matrices creates synergistic effects that significantly enhance therapeutic efficacy against wound infections [212]. Exemplifying this approach, hydrogels fabricated from gelatin/polyvinyl alcohol integrated with graphene oxide/silver nanoconjugates demonstrate not only potent bacterial growth inhibition but also excellent structural stability, highlighting their considerable potential for advanced wound management and biomedical applications [212]. Furthermore, hydrogels consisting of gelatin/polyvinyl alcohol and graphene oxide/silver nanoconjugates not only effectively inhibit bacterial growth but also maintain structural stability, suggesting their extensive potential applications in wound healing and biomedical fields [213]. Another innovative antibacterial hydrogel, which incorporates bacterial cellulose and polyvinyl alcohol (PVA) along with polyphenols and ε-poly-L-lysine, exhibits strong adhesion, mechanical performance, and antibacterial activity [214]. These hydrogels, through the integration of AMPs and functionalized nanomaterials, enhance their efficacy in infection control.

Light-sensitive and thermal-sensitive hydrogels present cutting-edge therapeutic solutions. For example, light-curable hydrogels, which include those with hyaluronic acid, possess antibiofilm and anti-inflammatory properties, demonstrating significant potential in the treatment of infected wounds [215]. Thermal-sensitive hydrogels, such as modified hydroxylbutyl chitosan (eLHBC) hydrogels, improve tissue adhesion, antimicrobial properties, and wound healing efficacy by encapsulating BMSCs [216]. Furthermore, light-triggered therapeutic diagnostic hydrogels, which incorporate AMP ε-poly-L-lysine (ePL) with photosensitizers, offer on-demand antimicrobial treatment and visual wound imaging capabilities [217]. Another example involves thermal-sensitive hydrogels that combine vB_AbaM-IME-AB2 phage and polymyxin, effectively managing *Acinetobacter baumannii* infections [218]. These hydrogels respond to specific stimuli, providing customized treatment approaches.

Supramolecular and self-assembling hydrogels hold innovative potential for enhancing hydrogel performance. For example, supramolecular hydrogels derived from PDGF-BB stimulate the growth of vascular endothelial cells, promote angiogenesis, and increase collagen deposition, suggesting their potential in chronic wound healing applications [219]. Self-assembling hydrogels made from ultrashort peptides, such as those combining FKF with gelatin, exhibit improved antibacterial properties, rendering them appropriate for treating skin infections and facilitating wound regeneration [220]. Additionally, hydrogels based on AMPs, like RWPIL, show promise in combating multidrug-resistant bacteria and accelerating wound healing [221]. These hydrogels leverage molecular self-assembly or supramolecular structures to enhance efficacy in the treatment of complex wounds.

Specialized functional hydrogels are adept at fulfilling specific therapeutic requirements. For example, layer-by-layer scaffolds (SL-B-L) deliver PDGF-BB and chlorhexidine to diabetic rat wounds, which not only enhances angiogenesis and the availability of collagen but also fosters interactions between keratinocytes and fibroblasts, while simultaneously reducing inflammation and levels of MMP-9 [222]. Furthermore, a photochemically cross-linked antimicrobial collagen hydrogel, enriched with nisin and levofloxacin, promotes antimicrobial effects and hastens the healing of infected wounds [223]. Through their tailored functional designs and therapeutic strategies, these hydrogels offer effective solutions for the innovative management of wounds.

In summary, AMP-based hydrogels exhibit a wide array of applications and groundbreaking potential within the realm of wound care. Ranging from self-healing and multifunctional hydrogels to those responsive to light and temperature, each variety of hydrogel fulfills a pivotal role in tackling various dimensions of wound management, including antimicrobial approaches, tissue regeneration, and therapeutic monitoring.

### 4.2. Nanomaterials

Recent breakthroughs in AMP-based Nanomaterials have shown significant promise for antibacterial applications. For example, nanocomplexes loaded with magnolol exhibit strong antibacterial and anti-inflammatory effects, significantly accelerating wound healing in murine models infected with *Staphylococcus aureus* [224]. Furthermore, mussel-inspired composite membranes that incorporate gold nanoparticles and AMPs have been found to enhance wound healing due to their synergistic blend of electroactivity, antibacterial potency, and antioxidative characteristics [225]. Gold nanorods functionalized with AMPs and subjected to near-infrared photothermal therapy have also exhibited enhanced wound healing in infected mice, with reduced cytotoxicity to human cells [226].

Furthermore, hydrogels embedded with peptide-loaded hollow silica nanoparticles and zinc alginate exhibit a broad-spectrum antimicrobial effect, which promotes rapid healing of chronic and MRSA-infected skin wounds [227]. Hydrogels based on solid lipid nanoparticles and lacticin 3147 exhibit enhanced antimicrobial properties against antibiotic-resistant bacteria, presenting promising avenues for the management of chronic wound infections [228]. Silver/silver chloride nanoparticles, engineered by *Bacillus* sp. protease, show promise as antimicrobial wound dressings by preserving protease activity and preventing absorption into the skin [229]. Additionally, wound films derived from silk cocoons and infused with silver nanoparticles exhibit potent antibacterial effects and expedite wound closure, highlighting their efficacy as innovative wound dressings [230]. Polymeric membranes mimicking the extracellular matrix, when enriched with silver nanoparticles, also unveil substantial potential for wound healing and antimicrobial action against Gram-positive bacteria [231]. Bilayered composite nanosheets, featuring wet-adhesive, hemostatic, and antimicrobial properties, effectively seal and heal soft-tissue bleeding wounds, showcasing their robust mechanical strength and bactericidal effectiveness [232].

To enhance wound healing and tissue regeneration, AMP-based nanoparticles have exhibited remarkable efficacy. For instance, a novel nanoconjugate combining apamin and ceftriaxone has been shown to expedite diabetic wound healing by enhancing tissue regeneration, mitigating inflammation, and fostering angiogenesis [233]. Nanoparticles composed of carboxymethyl chitosan, when loaded with the bioactive peptide OH-CATH30, exhibit a sustained release of OH30. This sustained release is instrumental in promoting cell migration and accelerating wound healing by improving granulation tissue formation and modulating cytokine expression [234]. Additionally, functionalized PVA-silk blended nanofibrous mats, which incorporate growth factors and the LL-37 peptide, facilitate diabetic wound healing by regulating the extracellular matrix and encouraging tissue remodeling [235].

Electrospinning technology has significantly propelled the creation of drug delivery systems embedded within nanofibers [236]. For example, the delivery of peptide/microRNA mixtures through electrospun nanofibrous wound dressings accelerates the healing process and improves the thickness of the epidermis and vascularization [237]. Additionally, electrospun nanofibrous mats containing nanoparticles and microparticles loaded with bioactive peptides exhibit superior wound healing capabilities, offering a promising solution for diabetic wounds [238].

These studies collectively underscore the multifaceted functionalities and promise of AMP-based nanoparticles in applications ranging from antibacterial treatments to wound healing and drug delivery. The progression of these nanomaterials presents innovative solutions for wound care and treatment, heralding potential future clinical applications.

### 4.3. Other Peptide Formulations

Decellularized placental sponges have shown significant antibacterial efficacy against extensively drug-resistant (XDR) clinical isolates, highlighting their potential as effective skin substitutes [239]. Specifically, human placental sponges, when infused with the AMP CM11 at a concentration of 64 µg/mL, display potent antibacterial properties and excellent biocompatibility, suggesting their efficacy in the fight against XDR bacteria [240]. Moreover, platelet-rich fibrin (PRF) membranes have been proven to improve wound healing and microbial control in both infected and non-infected wounds by decreasing inflammation and reducing wound size [240]. Antimicrobial and wound healing surfaces, modified with basic *Amino Acids*, have emerged as promising pH-monitorable wound dressings, providing protection against infections and aiding the healing process [241]. Glycerol and AMP-modified natural latex films (NRL-GI-AMP) exhibit sustained release of AMPs, excellent biocompatibility, and effective antibacterial activity, which greatly contribute to the healing of infected wounds [242]. Acellular dermal matrices adorned with collagen-affinity peptide (C-Histatin-1) expedite diabetic wound healing by continuously releasing Histatin-1, thereby enhancing angiogenesis and collagen deposition in diabetic ulcers [243]. Polyurethane wound dressings immobilized with LL37 peptide (PU-adhesive-LL37 dressing) exhibit potent antimicrobial effects against a variety of bacterial strains, while also promoting wound healing and re-epithelialization [244]. Additionally, composite membranes that incorporate recombinant silkworm AMP and poly(L-lactic acid) (PLLA) exhibit broad-spectrum antibacterial activity and selective cytotoxicity against cancer cells, underscoring their potential in cancer therapy and wound healing [245].

Emerging continuously are innovative antimicrobial strategies. The amalgamation of Synoeca-MP AMP with IDR-1018 amplifies proliferation, migration, and the expression of pro-regenerative genes within human skin cell cultures [246]. Exosomes, engineered to contain cathelicidin/LL-37, exhibit heightened antibacterial efficacy, foster endothelial cell tube formation, and stimulate the proliferation and migration of skin cells [247]. Antimicrobial ionic liquids, covalently conjugated with AMPs through “click” chemistry, maintain their efficacy against multidrug-resistant Gram-negative bacteria and display antibiofilm properties against *Klebsiella pneumoniae* [248]. PhaNP@Syn71, a bacteriophage-mimicking nanoparticle that incorporates silver-coated gold nanospheres and AMP Syn-71, effectively combats antibiotic-resistant *Streptococcus pyogenes*, facilitating infection-free wound healing and consistently reducing wound size [249].

Collagenase ointment, when used in conjunction with polymyxin B sulfate/bacitracin powder, accelerates the cleaning and healing of partial-thickness burn wounds. This combination reduces the time required to achieve a clean wound bed and promotes faster healing compared to silver sulfadiazine cream [250]. The CSBD-bFGF@CS/P5S9K composite dressing, which incorporates chitosan-binding bFGF (CSBD-bFGF) and the AMP P5S9K, exhibits potent antibacterial activity and expedites wound healing [251]. PLGA/Os-DA/IGF1-DA electrospun fibrous scaffolds, which integrate insulin-like growth factor 1 (IGF1) and the AMP Os, exhibit high hydrophilicity and antimicrobial properties, thereby supporting wound recovery and tissue regeneration [252]. High- and low-molecular-weight hyaluronan hybrid cooperative complexes (HCC) modulate inflammation biomarkers, metalloproteinases, and elastin, potentially increasing AMP expression and exhibiting enhanced wound healing capabilities [253].

Furthermore, “smart” polymer-based film-forming wound care formulations (FTP) incorporating pH-degradable and temperature-responsive polyacetal smart gels provide prolonged broad-spectrum antimicrobial protection [254]. Composite electrospun membranes with superhydrophobic hierarchical fiber/bead structures exhibit multifunctional capabilities, including antibacterial activity and enhanced wound healing for burn care applications [255]. Topically applied AMPs and hyaluronic acid compound masks improve skin barrier function and reduce adverse effects following ablative fractional carbon dioxide laser resurfacing [256]. Recombinant spider silk, functionalized with silkworm silk matrices, enhances cell adhesion, antimicrobial activity, and growth factor stimulation, presenting promising bioactive wound dressing and skin graft materials [257].

In addition to the aforementioned novel delivery systems, several classical antimicrobial peptide formulations and their combination regimens have demonstrated established value in clinical practice, providing a solid practical foundation for AMP-based therapeutic strategies [258]. A successful paradigm is the combined application of polymyxin B/bacitracin with collagenase. This regimen integrates active components with distinct mechanisms: polymyxin B and bacitracin, as classical antibiotic peptides, synergistically provide a broad antimicrobial spectrum covering both Gram-negative and Gram-positive bacteria, establishing a crucial local immune defense line for the wound [259,260]. Concurrently, collagenase achieves chemical debridement of the wound through specific degradation of devitalized necrotic tissue and denatured collagen, effectively dismantling the physical barrier that facilitates bacterial biofilm formation. This synergistic interaction not only efficiently cleanses the wound but also removes obstacles for subsequent formation of healthy granulation tissue and re-epithelialization, creating a favorable regenerative microenvironment [261,262]. Consequently, it significantly accelerates the healing process of full-thickness skin injuries and burn wounds [263]. This classic case provides important insights for developing AMP-based combination therapies that target multiple aspects of the wound healing cascade.

Triple-antibiotic ointment (TAO), which comprises neomycin, polymyxin B, and bacitracin, is highly effective in preventing infections in minor skin injuries. It boasts stable susceptibility profiles and presents a low risk of allergic reactions, making it a reliable choice for wound cleansing and infection prevention [264]. Polymyxin B sulfate-bacitracin zinc-neomycin sulfate ointment markedly diminishes scarring in comparison to gauze dressings, especially in reducing pigmentary changes [265]. The Polymyxin B sulfate-bacitracin powder ointment more effectively expedites the cleaning and healing of partial-thickness burn wounds than silver sulfadiazine cream [250]. Trolamine emulsion fosters quicker wound healing when contrasted with manuka honey or polymyxin-bacitracin ointment [266].

Matrices composed of chitosan, which are infused with innovative AMPs ASP-1 and ASP-2, have exhibited the capability to completely eliminate biofilms formed by multidrug-resistant bacteria that commonly infect wounds. These matrices release between 70 and 80% of the peptides over a period of 7 days in both in vitro and ex vivo conditions, highlighting their promise as effective topical treatments [267]. Ongoing research should persist in unraveling the mechanisms and exploring the clinical uses of these cutting-edge materials to further improve the efficacy of wound management.

## 5. Prospects and Perspectives

Peptide-based therapeutics have attracted considerable interest in recent years for their potential to revolutionize wound healing treatments. As the field continues to propel, numerous promising avenues and innovations are emerging, with the potential to fundamentally transform the landscape of wound care. The application of new technologies and the development of novel materials are driving revolutionary changes in wound treatment methods, significantly enhancing therapeutic outcomes and patient prognosis.

Despite their promising clinical prospects, the translational development of AMPs faces multiple challenges [258]. In addition to issues such as poor stability, potential cytotoxicity, and high production costs, the emergence of bacterial resistance remains a significant concern [258,268]. Studies indicate that bacteria can employ various mechanisms to counteract AMPs, including modifications of membrane charge, activation of efflux pumps, secretion of proteases, and formation of biofilms [268]. However, compared to conventional antibiotics, AMPs exert their effects by targeting the fundamental structure of bacterial membranes and through multi-target mechanisms, which substantially raise the genetic barrier for the development of resistance [269]. This unique attribute preserves their distinct advantage in combating multidrug-resistant bacterial infections [258]. Future research should focus on rational structural optimization and advanced delivery systems to enhance their stability and safety while maximizing this inherent resistance-reluctant property [258].

The innovative design of AMPs’ drug delivery and release systems enhances the stability of AMPs while achieving precise targeted delivery and sustained release. AMPs exhibit significant potential when integrated with new technologies, such as nanotechnology, supramolecular assembly, biodegradable carriers, and targeted delivery systems. In nanotechnology, combining AMPs with nanomaterials has led to the creation of nanocarriers with improved stability, targeted delivery, and prolonged release capabilities, thus improving wound healing outcomes [270]. Functionalization and conjugation strategies have further broadened the scope of these nanocarriers in drug discovery and delivery systems [271]. Supramolecular assembly offers a promising method to enhance AMP stability and efficacy in wound treatment. For example, AMP F2I-LL self-assembles into a nanofiber-structured hydrogel in a simulated physiological environment, showing broad-spectrum antimicrobial activity. This self-assembly approach improves bactericidal efficacy and enhances serum stability compared to monomeric peptides [272]. Biodegradable carriers present another method for delivering AMPs. These carriers enable controlled release, extending the antimicrobial activity of AMPs and supporting tissue repair. For instance, a novel biodegradable dressing made of an alginate membrane and polycaprolactone nanoparticles loaded with curcumin exhibits multifunctional properties crucial for wound healing, including high adherence, controlled drug release, and permeability regulation [273]. Targeted delivery technologies allow precise administration of AMPs to specific wound sites, minimizing systemic side effects and enhancing local therapeutic effects [274].

The development and optimization of AMP technologies have significantly enhanced their antimicrobial efficacy and biological stability through chemical synthesis and genetic engineering. Recent progress includes the use of site-directed mutagenesis, truncation, hybridization, capping, and cyclization to optimize AMP properties, with solid-phase peptide synthesis facilitating their applications in agriculture, animal husbandry, the food industry, and medicine [275]. Breakthroughs in high-throughput screening technologies, along with chemical modifications and structural constraints, have improved AMP binding affinity, selectivity, membrane permeability, and metabolic stability, providing a solid foundation for the next generation of therapeutic AMPs [276]. Additionally, genetic engineering offers a more cost-effective solution, expanding AMP applications in the medical field [277].

The integration of AMPs with emerging technologies is driving innovative breakthroughs in medicine. Significant progress includes a novel smart bandage that uses electrochemical technology to detect bacterial virulence factors in wounds with high sensitivity [278]. AMPs are also used in smart wound dressings that automatically sense changes in the wound environment and release antimicrobial agents to accelerate healing [279]. Incorporating AMPs into tissue engineering scaffolds provides antimicrobial protection and supports cellular proliferation, with collagen valued for its biocompatibility and low antigenicity [280]. Additionally, gene-activated scaffolds, such as those incorporating cell-penetrating peptides to deliver pSDF-1α, enhance angiogenesis [281].

Combining AMPs with natural products and conventional antibiotics is an effective strategy for enhancing wound healing and combating antibiotic resistance. Research shows that incorporating AMPs with plant extracts and propolis into chitosan hydrogels significantly improves wound healing outcomes [282]. Additionally, combining conventional antibiotics with peptides tagged with non-natural *Amino Acids* can enhance antimicrobial activity and markedly reduce lipopolysaccharide release from Gram-negative bacteria [283]. Personalized AMP therapies, tailored to patient genomics, wound characteristics, and pathogen profiles, aim to improve treatment precision, while structural modifications and nanoparticle integration seek to enhance stability and effectiveness. Despite promising results, further research is needed to assess the environmental and human health impacts of these novel AMPs [284]. Moreover, the efficacy of AMPs in treating inflammatory skin diseases and improving healing in infected wounds has been validated, highlighting their potential in translational medicine [285].

The application of AMPs in specific wounds and diseases exhibits significant potential. In treating diabetic ulcers, the optimized RL-QN15 peptide combined with Zn^2+^-crosslinked sodium alginate hydrogel can accelerate cell proliferation, migration, and angiogenesis while reducing inflammation [286]. In tumor-related wound healing, AMPs are emerging as alternatives to conventional antibiotics and chemotherapy, especially for multidrug-resistant pathogens and drug-resistant cancers. Encapsulation in nanocarriers improves AMP stability and delivery efficiency; for example, encapsulating AMP NRC-07 in chitosan nanoparticles enhances its antimicrobial activity and exhibits selective cytotoxicity against cancer cell lines [287]. Additionally, AMPs are gaining attention for skin grafting applications, including infection prevention, graft integration, and accelerated wound healing, suggesting they could play a significant role in clinical settings in the future [288].

In maintaining microbial diversity within wounds, defensins derived from keratinocytes help prevent skin dysbiosis and bacterial infections by activating specific receptors on neutrophils. These defensins play a crucial role in combating *Staphylococcus aureus* infections and maintaining a healthy microbial ecosystem on the skin. Research indicates that this signaling axis is essential for preserving microbial diversity and host defense [289]. In the area of antioxidant modification, phosphorylated collagen peptides derived from fish bones exhibit enhanced calcium-binding capacity and stability, which is beneficial for wound healing and bioactivity enhancement. These modified peptides also exhibit significant antioxidant properties, making them promising candidates for supplements and therapeutic applications [290]. Furthermore, research into AMPs for nerve repair reveals their potential in treating peripheral nerve damage, particularly in regulating nerve innervation and promoting nerve regeneration [291,292].

The stability of AMPs during low-temperature storage and transportation is crucial for their clinical applications. Recent progress has focused on improving AMP stability through innovative preservation methods, notably using antifreeze peptides to control ice crystal growth. One such peptide, AVD, demonstrates excellent recrystallization inhibition, solubility, and biocompatibility, making it an ideal cryoprotectant. Molecular dynamics simulations and mutation analysis reveal that residues Thr6 and Asn8 are critical for the peptide’s cryoprotective ability, while residue Ser18 enhances its interaction with ice. The successful application of AVD in cell cryopreservation shows high post-freezing recovery rates, providing new insights and opportunities for enhancing AMP stability in clinical settings [293].

## 6. Conclusions

In conclusion, peptide-based therapeutics represent a cutting-edge progress in the field of wound healing, providing targeted, efficient, and innovative treatment solutions. Ongoing research and technological progress suggest that these therapeutics could revolutionize wound management, enhance patient recovery, and alleviate the challenges posed by chronic and recalcitrant wounds. Sustained investment and exploration in this domain are crucial for unlocking the full potential of peptide-based approaches to wound healing.

## Figures and Tables

**Figure 1 biomolecules-15-01613-f001:**
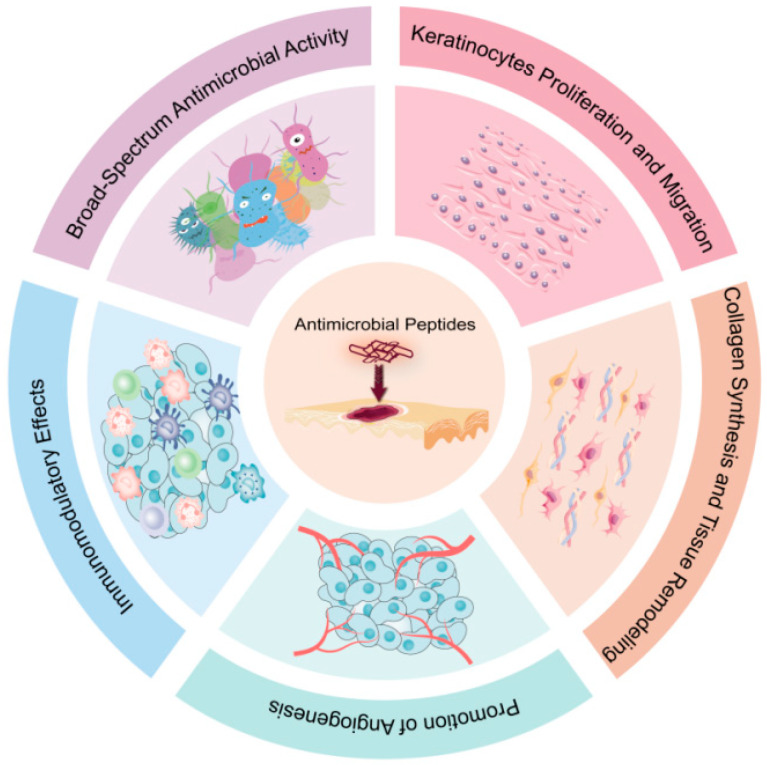
Mechanisms of action of AMPs in Skin Wound Healing. AMPs play a crucial role in skin wound healing through multiple mechanisms: keratinocytes migration and proliferation; collagen synthesis and tissue remodeling; promotion of angiogenesis; immunomodulatory effects; broad-spectrum antimicrobial activity.

**Figure 2 biomolecules-15-01613-f002:**
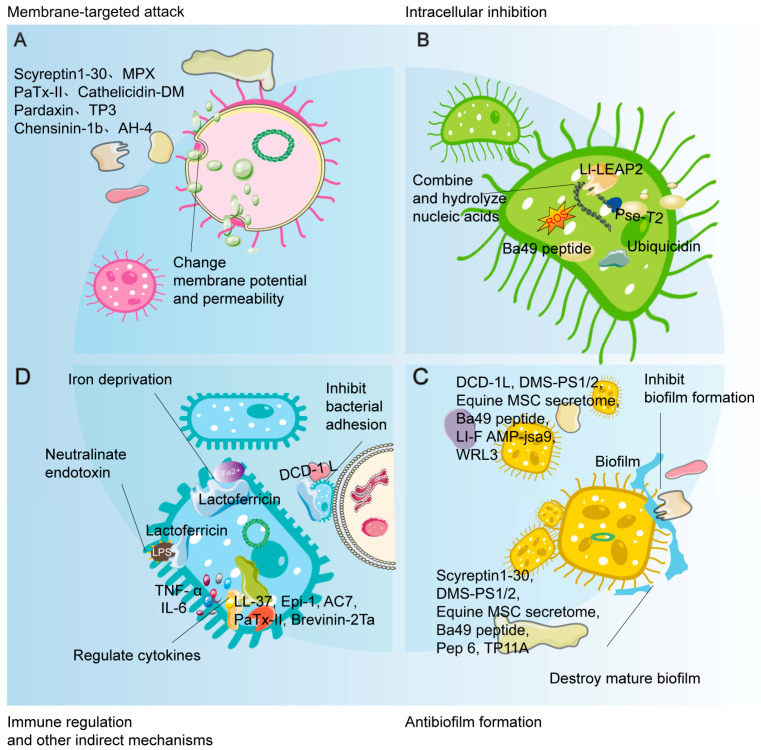
Schematic illustration of the multi-mechanistic and synergistic antimicrobial actions of AMPs. This figure systematically summarizes the broad-spectrum antimicrobial activity of AMPs through four major mechanisms: (**A**) Membrane-targeted attack: disrupting membrane integrity or altering membrane potential, leading to leakage of cellular contents; (**B**) Intracellular inhibition: upon entering the cell, exerting multi-target effects by binding nucleic acids, interfering with physiological processes, and inducing reactive oxygen species (ROS); (**C**) Anti-biofilm activity: effectively inhibiting biofilm formation and disrupting established mature biofilms; (**D**) Immunomodulatory and other indirect mechanisms: indirectly enhancing host defense by neutralizing endotoxins (e.g., LPS), modulating cytokines, and sequestering iron ions. Collectively, these mechanisms constitute a multi-dimensional, synergistic network crucial for their potent antimicrobial efficacy and low propensity for inducing resistance.

**Figure 3 biomolecules-15-01613-f003:**
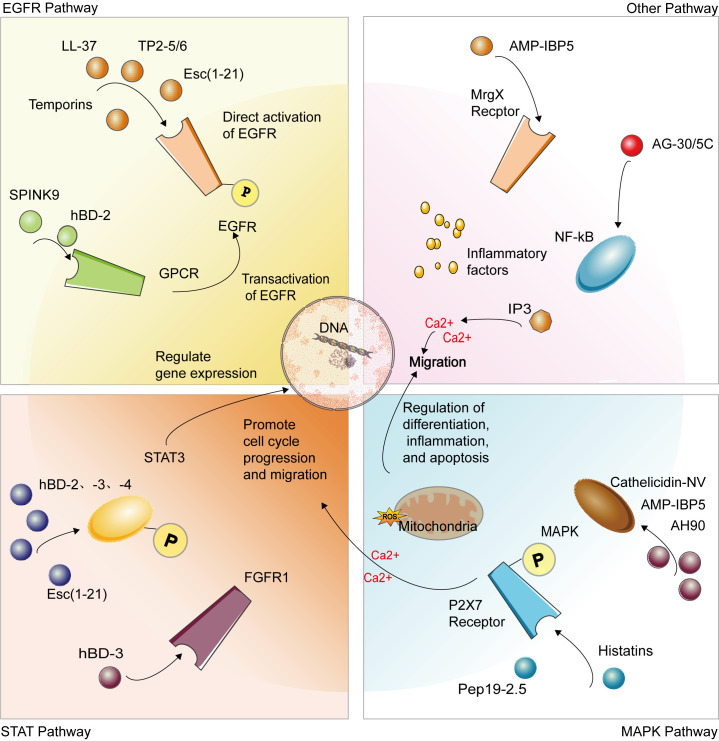
AMPs coordinate key processes in skin wound healing by activating a complex signaling network in keratinocytes. The primary mechanisms include: direct or indirect activation of core signaling pathways such as EGFR, STAT3, MAPK (ERK1/2, JNK, p38), and NF-κB; triggering secondary messenger events, including the mobilization of intracellular calcium ions (Ca^2+^) and induction of mitochondrial reactive oxygen species (ROS) production; and converging ultimately on enhanced core cellular functions of keratinocytes—promoting migration and proliferation, inhibiting apoptosis, and modulating the secretion of inflammatory factors. These synergistic actions collectively accelerate re-epithelialization, facilitate wound closure, and coordinate the immune microenvironment, representing the crucial molecular basis through which AMPs repair skin damage.

## Data Availability

No new data were created or analyzed in this study.

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
