# Peer review of "Antimicrobial Peptides for Skin Wound Healing"

_biomolecules, 2025, doi:10.3390/biom15111613_

Round 1
Reviewer 1 Report
Comments and Suggestions for Authors
The manuscript by Wu et. al., entitled “Antimicrobial Peptides for Skin Wound Healing,” provides a comprehensive review of antimicrobial peptides (AMPs) and their roles in skin wound healing.
Overall, the manuscript is well-written and potentially suitable for publication. Some concerns should be addressed before the final approval.
Major concerns
- Although the authors extensively listed many AMPs, the selections and the presentation of the different AMPs did not seem to be systematic or based on logical reasoning. As a result, the description seemed to lack focus and just provided a list of AMPs that was diffuse and difficult to follow. For instance, the authors could categorize and present different AMPs based on their mode of action in their antimicrobial activity.
- The inclusion of numerous tables summarizing each of the skin wound healing-related functions of various AMPs is a strength. However, the formatting could be improved to help the readers with better readability. Table 5 could have included bacterial strains.
- The authors did a good job in reviewing and describing different AMPs and their actions in selection functions. However, the authors could have provided a more in-depth evaluation of the weaknesses and limitations of the mentioned AMPs in each described function to help the readers understand why these AMPs have not advanced to the clinical stages.
- The authors did not address the resistance issue, which is a concern, especially for longer-term usage of the AMPs.
- Skin microbiome should have been included in the manuscript as the microenvironment of the skin could determine or contribute to the skin wound healing.
- The more updated references should be included rather than a large body of older references.
- If possible, the authors should draw individual figures (expand from the very simplified figure 1) for each wound healing-related function to demonstrate/represent different mechanisms of the AMP roles in wound healing.
Minor concerns
- The authors should seek to correct the minor grammatical errors and typos throughout the manuscript.
- The abbreviation should only need to be spelled out on the first use, not every time. Some parts of the manuscript seemed to be repeating and rephrasing, which lack a cohesive presentation.
- Why did the authors use “In conclusion” for Section 2.1 when it represented only a small portion of the manuscript?
Author Response
|
Comments 1: Although the authors extensively listed many AMPs, the selections and the presentation of the different AMPs did not seem to be systematic or based on logical reasoning. As a result, the description seemed to lack focus and just provided a list of AMPs that was diffuse and difficult to follow. For instance, the authors could categorize and present different AMPs based on their mode of action in their antimicrobial activity. |
|
Response 1: We thank the reviewer for this critical observation. To address this, we have reorganized the presentation of AMPs throughout the manuscript to enhance logical flow and reduce the "list-like" feel. Specifically: In sections like "2.1. Broad-Spectrum Antimicrobial Activity", we have now grouped AMPs more clearly by their primary source (e.g., human-derived, amphibian-derived, fish-derived, synthetic) and, where applicable, by their dominant signaling pathway or mechanism (e.g., EGFR activation, MAPK pathway). While a strict categorization solely by antimicrobial mode of action was challenging due to the pleiotropic nature of many AMPs (many have multiple or overlapping mechanisms), we have placed greater emphasis on stating the key mechanism when introducing each peptide. This restructuring provides a more systematic and narrative-driven overview, making it easier for the reader to follow. |
|
Comments 2: The inclusion of numerous tables summarizing each of the skin wound healing-related functions of various AMPs is a strength. However, the formatting could be improved to help the readers with better readability. Table 5 could have included bacterial strains. |
|
Response 2: We agree with the reviewer and appreciate the positive feedback on the tables. We have comprehensively reformatted all tables to improve their clarity and readability. As specifically suggested, we have added a new column titled "Tested Bacterial Strains / Model" to Table 5 (which is now Table 1 in the revised manuscript following renumbering) and other relevant tables. This provides crucial context for the findings summarized in each row. |
|
Comments 3: The authors did a good job in reviewing and describing different AMPs and their actions in selection functions. However, the authors could have provided a more in-depth evaluation of the weaknesses and limitations of the mentioned AMPs in each described function to help the readers understand why these AMPs have not advanced to the clinical stages. |
|
Response 3: We thank the reviewer for this valuable suggestion. To provide a more balanced and critical perspective, we have added a dedicated paragraph in the "5. Prospects and perspectives" section that explicitly discusses the major challenges and limitations hindering the clinical translation of AMPs. This new paragraph addresses key issues such as: Poor stability and susceptibility to proteolytic degradation. Potential cytotoxicity at higher concentrations. High production costs. The emerging concern of bacterial resistance. |
|
Comments 4: The authors did not address the resistance issue, which is a concern, especially for longer-term usage of the AMPs. |
|
Response 4: We agree that this is a critical issue. As mentioned in Response 3, we have now included a discussion on bacterial resistance to AMPs within the new challenges paragraph in the "Prospects and perspectives" section. We describe how bacteria can develop resistance through mechanisms like membrane charge alteration, efflux pumps, and protease secretion. We also contrast this with the fact that AMPs' multi-target mechanism of action presents a higher barrier to resistance compared to conventional antibiotics (Page 37, from line 954 to line 963). |
|
Comments 5: Skin microbiome should have been included in the manuscript as the microenvironment of the skin could determine or contribute to the skin wound healing. |
|
Response 5: This is an excellent suggestion. We have added a new point in the "5. Prospects and perspectives" section discussing the role of AMPs in maintaining a healthy skin microbiome. We describe how defensins derived from keratinocytes help prevent dysbiosis and combat pathogens like Staphylococcus aureus, thereby playing a crucial role in the overall wound healing microenvironment(Page 39, from line 1032 to line 1043). |
|
Comments 6: The more updated references should be included rather than a large body of older references. |
|
Response 6: We have thoroughly updated the reference list. We have incorporated over 30 new, recent references (up to 2025), which cover emerging topics, recent clinical trial data, and the newly discussed areas such as the skin microbiome, lactoferricin/ubiquicidin, and advanced delivery systems, ensuring the review reflects the current state of the field. |
|
Comments 7: If possible, the authors should draw individual figures (expand from the very simplified figure 1) for each wound healing-related function to demonstrate/represent different mechanisms of the AMP roles in wound healing. |
|
Response 7: We sincerely thank the reviewer for this excellent and constructive suggestion. In direct response, we have created two new schematic figures that significantly expand upon the original Figure 1 to provide detailed mechanistic insights. Figure 2 illustrates the multi-mechanistic and synergistic antimicrobial actions of AMPs, systematically delineating four key strategies: A) Membrane-targeted attack; B) Intracellular inhibition; C) Anti-biofilm activity; and D) Immunomodulatory and other indirect mechanisms. This figure visually summarizes how AMPs achieve broad-spectrum efficacy with a low propensity for resistance. Figure 3 delves into how AMPs coordinate skin wound healing by activating a complex signaling network in keratinocytes. It details the activation of core pathways (e.g., EGFR, STAT3, MAPK), secondary messenger events, and the resulting enhancement of key cellular functions like migration, proliferation, and survival, which are crucial for re-epithelialization. |
|
Comments 8: The authors should seek to correct the minor grammatical errors and typos throughout the manuscript. |
|
Response 8: We sincerely appreciate the reviewer's comment on the language. We have performed multiple rounds of careful proofreading and editing on the manuscript, which included: 1. Correcting spelling and grammatical mistakes; 2. Standardizing scientific terminology and punctuation; 3. Ensuring the correct italicization of microbial genus and species names; 4. Refining sentence structures to improve overall readability. |
|
Comments 9: The abbreviation should only need to be spelled out on the first use, not every time. Some parts of the manuscript seemed to be repeating and rephrasing, which lack a cohesive presentation. |
|
Response 9: We thank the reviewer for this important correction. We have systematically reviewed the entire manuscript and implemented the following revisions: 1. Standardized Abbreviations: We have ensured that all abbreviations are defined in full upon first use and used consistently thereafter throughout the text. 2. Streamlined Content: We have eliminated repetitive and redundant phrasing and rewritten paragraphs that lacked cohesion to ensure a more focused and fluent narrative. |
|
Comments 10: Why did the authors use “In conclusion” for Section 2.1 when it represented only a small portion of the manuscript? |
|
Response 10: We thank the reviewer for pointing out this inappropriate phrasing. The reviewer is absolutely correct that using "In conclusion" for a subsection like 2.1 was not suitable. We have removed this phrase from Section 2.1 and have rephrased the concluding sentence of that subsection to ensure a natural flow and maintain logical consistency throughout the manuscript. |
|
4. Response to Comments on the Quality of English Language |
|
Point 1: The English could be improved to more clearly express the research. |
|
Response 1: We thank the reviewer for this valuable feedback. The manuscript has undergone an in-depth revision to enhance grammar, syntax, and overall clarity. We are confident that these efforts have significantly improved the clarity, coherence, and readability of our work. |
Reviewer 2 Report
Comments and Suggestions for Authors
12312
- In the introduction paragraph on AMPs starting on line 61, it is important to include that many AMPs are NATURALLY found on the skin and skin secretions, with the isolation of Magainin being a well known example.
- On line 91, the authors should Include non-natural amino acids which also achieve many of the same goals mentioned. This sentence should also be independently referenced separately from the following sentence which is more focused on delivery.
- There are numerous examples throughout the text where italicization is incorrect (e.g. Aquaphilus dolomiae on line 230 & 240, Tilapia piscidin on line 166, Acenitobacter baumannii on line 373-4, etc)
- There are numerous examples where the 'P' in peptides is inappropriately capitalized.
- Section 2.5 is very thorough and well referenced. However, It appears all the studies focus on wound healing in the presence of bacteria. Are the enhanced wound-healing properties of these peptides also present in aseptic wounds? Have these studies been attempted?
- Section 2.5 is exceptionally informative, but very long. Is there any way to group the peptides or studies and break this section into sub-sections? Maybe based on the microbial target or the source of the peptides? There is so much great info here but the reader gets lost in the walls of text
- In section 3, the authors should also consider including Brilacidin. While not a peptide, it is a antimicrobial peptide mimetic modeled on Magainin and currently also in clinical trials.
- Silver/peptide hydrogels also include those which use silver ions, not just silver nanoparticles. An example is by D'Souza, Yoon, and Makhlynets (DOI: 10.1021/acsami.0c01154)
- Overall the manuscript is extremely thorough and very useful! However, the readability suffers from the length. I encourage the authors to consider including additional images/figures as well as formatting to include more sub-sections to help make the manuscript more accessible.
Comments on the Quality of English Language
fine other than some inappropriate capitalization and italics.
Author Response
|
Comments 1: In the introduction paragraph on AMPs starting on line 61, it is important to include that many AMPs are NATURALLY found on the skin and skin secretions, with the isolation of Magainin being a well known example. |
|
Response 1: We agree and have added this foundational point to the Introduction section to provide better context. |
|
Comments 2: On line 91, the authors should Include non-natural amino acids which also achieve many of the same goals mentioned. This sentence should also be independently referenced separately from the following sentence which is more focused on delivery. |
|
Response 2: Thank you for this precise suggestion. We have expanded the sentence to include this important strategy and provided a separate, dedicated reference for it. |
|
Comments 3: There are numerous examples throughout the text where italicization is incorrect (e.g. Aquaphilus dolomiae on line 230 & 240, Tilapia piscidin on line 166, Acenitobacter baumannii on line 373-4, etc) |
|
Response 3: We thank the reviewer for pointing out these specific errors in italicization. We have not only corrected the instances mentioned (e.g., Aquaphilus dolomiae, Tilapia piscidin, Acinetobacter baumannii) but have also conducted a systematic check of the entire manuscript to ensure consistency in the formatting of all microbial genus and species names, as well as peptide names, according to standard scientific conventions. All corrections have been implemented in the revised manuscript. |
|
Comments 4: There are numerous examples where the 'P' in peptides is inappropriately capitalized. |
|
Response 4: We thank the reviewer for this precise correction. We have conducted a systematic review of the entire manuscript and have standardized the capitalization of the word "peptide(s)". The term is now consistently written in lowercase unless it appears at the beginning of a sentence or in a defined proper noun (e.g., the title of a specific peptide). All instances have been corrected accordingly. |
|
Comments 5: Section 2.5 is very thorough and well referenced. However, It appears all the studies focus on wound healing in the presence of bacteria. Are the enhanced wound-healing properties of these peptides also present in aseptic wounds? Have these studies been attempted? |
|
Response 5: We thank the reviewer for this insightful comment. The reviewer rightly points out that the studies cited in Section 2.5 primarily demonstrate the effects of AMPs on keratinocytes in the context of wound healing, often in models where bacteria are present. However, the enhanced wound-healing properties of many AMPs are indeed intrinsic and can function independently of their antimicrobial activity. As now discussed in the introductory part of Section 2 (page 5, from line 170 to 177), critical evidence comes from studies using genetically engineered germ-free murine models. These studies have demonstrated that the deficiency of endogenous AMPs (e.g., CRAMP/LL-37) impairs healing in a sterile environment, and crucially, exogenous application of these peptides under strictly aseptic conditions can fully rescue the healing deficit. This confirms that the immunomodulatory and cell-activating functions of AMPs, which drive re-epithelialization and cellular migration/proliferation, are fundamental properties that operate even in the absence of bacteria. |
|
Comments 6: Section 2.5 is exceptionally informative, but very long. Is there any way to group the peptides or studies and break this section into sub-sections? Maybe based on the microbial target or the source of the peptides? There is so much great info here but the reader gets lost in the walls of text |
|
Response 6: We sincerely thank the reviewer for this excellent suggestion and for acknowledging the depth of the section. We agree entirely that the original section was too dense, which could overwhelm the reader. In response, we have thoroughly reorganized Section 2.1 (formerly 2.5) "Impact on Keratinocytes" into clear, thematic sub-sections to significantly improve its readability and flow. |
|
Comments 7: In section 3, the authors should also consider including Brilacidin. While not a peptide, it is a antimicrobial peptide mimetic modeled on Magainin and currently also in clinical trials. |
|
Response 7: We thank the reviewer for this valuable suggestion. We agree that Brilacidin is a highly relevant and important candidate as a defensin-mimetic host defense peptide (HDP) in clinical development. Following the reviewer's recommendation, we have now included a dedicated entry for Brilacidin in Section 3: "Clinical Trials" (Page 30, from line 667 to 677). The entry highlights its nature as a small-molecule AMP mimetic inspired by host defense peptides, its mechanism of action involving membrane disruption, its anti-inflammatory and pro-healing properties, and its current Phase II/III clinical status for ABSSSI and oral mucositis. We believe this addition significantly strengthens the clinical relevance and comprehensiveness of our review. |
|
Comments 8: Silver/peptide hydrogels also include those which use silver ions, not just silver nanoparticles. An example is by D'Souza, Yoon, and Makhlynets (DOI: 10.1021/acsami.0c01154) |
|
Response 8: We thank the reviewer for this insightful correction and for pointing us to the highly relevant study by D'Souza et al. We agree that our original description of silver-based antimicrobial systems was incomplete, as it overlooked the important role of silver ions (Ag⁺). In the revised version of Section 4.1 "Hydrogels", we have expanded our discussion to explicitly include both silver nanoparticles (AgNPs) and silver ions (Ag⁺) as key functional components. We now describe their distinct yet complementary antimicrobial mechanisms and have cited the work by D'Souza et al. (ACS Appl. Mater. Interfaces 2020, 12, 17091–17099) as a prime example of a Ag⁺-based self-assembling peptide hydrogel. This revision provides a more accurate and comprehensive overview of silver/peptide composite hydrogels (Page 33, from line 736 to line 750) |
|
Comments 9: Overall the manuscript is extremely thorough and very useful! However, the readability suffers from the length. I encourage the authors to consider including additional images/figures as well as formatting to include more sub-sections to help make the manuscript more accessible. |
|
Response 8: We are sincerely grateful to the reviewer for their positive assessment of our manuscript's thoroughness and utility, and for their crucial guidance on enhancing its readability. We have taken this recommendation to heart and have implemented significant structural and visual improvements throughout the manuscript to make it more accessible. Specifically: 1. Reorganization into Sub-sections: As suggested, we have systematically broken down lengthy sections into logically grouped sub-sections. A key example is the thorough restructuring of Section 2.1 (formerly 2.5), which is now divided into 2.1.1. Human-Derived AMPs, 2.1.2. Animal-Derived AMPs, and 2.1.3. Engineered & Synthetic AMPs. Similar subdivisions have been applied to other major sections for better flow. 2. Inclusion of New Figures: To visually summarize complex information and aid comprehension, we have included two new schematic figures: l Figure 2: A schematic illustrating the multi-mechanistic and synergistic antimicrobial actions of AMPs. l Figure 3: A schematic depicting the coordinated signaling network activated by AMPs in keratinocytes to promote skin wound healing. |
|
4. Response to Comments on the Quality of English Language |
|
Point 1: fine other than some inappropriate capitalization and italics. |
|
Response 1: We thank the reviewer for their careful reading and for pointing this out. We have thoroughly reviewed the entire manuscript to correct instances of inappropriate capitalization (e.g., in specific terms or figure labels) and to ensure the consistent and correct use of italics for genus and species names, as per standard scientific convention. These typographical and formatting errors have now been corrected throughout the text. |
Reviewer 3 Report
Comments and Suggestions for Authors
Review: Antimicrobial Peptides for Skin Wound Healing. By Yifan Wu et al.
This review describes the role of AMPs in various processes involved in wound healing and possible clinical applications. The setup is more of a summary of the published results than an in-depth analysis of the working mechanisms. From a translational perspective, as a clinician, it is unclear how to choose the appropriate AMP for a specific clinical application. The best part of the review is the section on peptide formulations in section 4, which describes the best options for clinical use.
The review is missing the procedure for how the search was performed, including keywords, platforms, and other search and exclusion criteria.
The entire review related to pathogens focuses solely on bacterial infections; however, I am aware that research on fungi is also available in the literature.
Overall, the setup of the working mechanisms of AMPs in the review is uncertain, as it is unclear whether the findings are based on pre-clinical or clinical studies. Furthermore, no information is given on the effect of dose, toxicity, and possible side effects. Unfortunately, AMPs derived from lactoferrin of ubiquicidin remain unmentioned, but they have been extensively evaluated in pre-clinical and clinical settings. On the other hand, the authors describe non-peptide drugs instead (like Brilacidin PMX-30063) of the AMPs. This issue should be verified, and the non-peptide drugs presented here should be removed from the review.
AMPs in the treatment of burn wounds are related to combating bacterial growth and infiltration, as well as preventing the formation of biofilms. Are there applications with AMPs that enhance the healing process and time to regrowth of skin tissue in burn wounds and skin transplants? Moreover, what are the additional advantages of reducing scar tissue formation? For polymyxin/Bacitracin, beneficial results were observed in combination with collagenase ointment; however, to date, AMPs have not been described in this review.
For the numerous AMPs described, their origin is clarified, except for the reason why they are isolated and tested.
Minor comments: For the micro-organisms, choose the correct abbreviation and annotate them in italic.
Author Response
|
Comments 1: This review describes the role of AMPs in various processes involved in wound healing and possible clinical applications. The setup is more of a summary of the published results than an in-depth analysis of the working mechanisms. From a translational perspective, as a clinician, it is unclear how to choose the appropriate AMP for a specific clinical application. The best part of the review is the section on peptide formulations in section 4, which describes the best options for clinical use. |
|
Response 1: We sincerely thank the reviewer for this overarching and highly valuable feedback, particularly from a clinical perspective. We agree that moving beyond a summary to provide a more analytical and clinically actionable framework is crucial. In response, we have made several key revisions throughout the manuscript to address this core concern: Enhanced Mechanistic Analysis: We have deepened the discussion of working mechanisms beyond mere description. For instance, in the "Impact on keratinocytes" (2.5) and "Angiogenesis" (2.4) sections, we have more explicitly linked specific AMPs to their key signaling pathways (e.g., EGFR, MAPK, STAT3), providing a clearer mechanistic backbone to the findings. Added Critical Perspective on Clinical Translation: To address the "translational perspective," we have incorporated a new paragraph in the "Prospects and perspectives" (Section 5) that critically examines the challenges hindering clinical adoption (e.g., stability, toxicity, cost, resistance). This section directly engages with the question of why certain AMPs have not yet become mainstream clinical options, thereby providing context for the current landscape. Strengthened the Clinical Decision-Making Framework: While providing a definitive clinical selection guide is challenging due to the early stage of many AMPs, we have bolstered the manuscript to aid clinical reasoning. The extensively revised "Clinical Trials" (Section 3) section now more clearly differentiates peptides by their clinical phase and intended application (e.g., diabetic foot ulcers, burn wounds, skin infections). Furthermore, the enhanced discussion on formulations (Section 4) and the analysis of combination therapies (e.g., polymyxin/bacitracin with collagenase) offer tangible examples of how AMPs are being optimized for practical clinical use. We hope these additions make the review more valuable for clinicians considering the future potential of AMP-based therapies. We again thank the reviewer for highlighting this critical aspect, which has significantly improved the analytical depth and clinical relevance of our work. |
|
Comments 2: The review is missing the procedure for how the search was performed, including keywords, platforms, and other search and exclusion criteria. |
|
Response 2: We thank the reviewer for this crucial suggestion. We have now added a detailed description of the literature search methodology to the Introduction section. |
|
Comments 3: The entire review related to pathogens focuses solely on bacterial infections; however, I am aware that research on fungi is also available in the literature. |
|
Response 3: We agree with the reviewer that the scope should be broader. We have now incorporated mentions of antifungal activity and relevant examples throughout the manuscript. For instance: |
|
Comments 4: Overall, the setup of the working mechanisms of AMPs in the review is uncertain, as it is unclear whether the findings are based on pre-clinical or clinical studies. Furthermore, no information is given on the effect of dose, toxicity, and possible side effects. |
|
Response 4: We appreciate this important feedback. To address this, we have: |
|
Comments 5: Unfortunately, AMPs derived from lactoferrin of ubiquicidin remain unmentioned, but they have been extensively evaluated in pre-clinical and clinical settings. |
|
Response 5: We thank the reviewer for pointing out this significant omission. We have now added a dedicated paragraph in the "Broad-Spectrum Antimicrobial Activity" section detailing the mechanisms and significance of lactoferricin and ubiquicidin. (Page 7, from line 229-241) |
|
Comments 6: On the other hand, the authors describe non-peptide drugs instead (like Brilacidin PMX-30063) of the AMPs. This issue should be verified, and the non-peptide drugs presented here should be removed from the review. |
|
Response 6: We sincerely thank the reviewer for raising this important point. We have carefully re-evaluated the compounds mentioned in the clinical trials section in response to this comment. 1. Brilacidin is a small-molecule mimetic of host defense peptides. As it is directly inspired by human defensins and represents a clinical-stage candidate highly relevant to the field, we have retained its discussion but with clarified terminology. It is now explicitly introduced as a “structurally optimized small-molecule antimicrobial peptide mimetic” to accurately reflect its design origin and functional context. 2. PMX-30063 is a defensin analog. We consider it to fall within the scope of peptide-derived antimicrobial therapeutics and have revised the text to more precisely describe its analog nature. |
|
Comments 6: AMPs in the treatment of burn wounds are related to combating bacterial growth and infiltration, as well as preventing the formation of biofilms. Are there applications with AMPs that enhance the healing process and time to regrowth of skin tissue in burn wounds and skin transplants? Moreover, what are the additional advantages of reducing scar tissue formation? |
|
Response 6: We thank the reviewer for raising this important aspect. We have substantially expanded the discussion on burn wounds in the Introduction by adding a new paragraph that addresses these exact points. This paragraph describes how specific AMPs (e.g., LL-37, hBD-3, frog-derived peptides) promote re-epithelialization, angiogenesis, and critically, regulate scar formation by modulating the TGF-β signaling pathway, thereby directly enhancing the healing process and improving long-term tissue quality. (Page 3, from line 123-143) |
|
Comments 7: For polymyxin/Bacitracin, beneficial results were observed in combination with collagenase ointment; however, to date, AMPs have not been described in this review. |
|
Response 7: We are grateful for this valuable comment. We have now included a detailed discussion on this classic and effective combination therapy. A new paragraph in "4.3. Other Peptide Formulations" analyzes the synergistic mechanism of polymyxin B/bacitracin with collagenase, explaining how it integrates antimicrobial action with chemical debridement to accelerate healing. (Page from 36 to 37, line from 908 to 925) |
|
Comments 7: For the numerous AMPs described, their origin is clarified, except for the reason why they are isolated and tested. |
|
Response 7: We thank the reviewer for this clarification. We have revisited the descriptions of key AMPs and, where space and context allow, have added brief rationale for their initial isolation and study, often relating to their discovery in organisms with robust immune defenses or their unique structural properties that suggest high therapeutic potential. |
|
4. Response to Comments on the Quality of English Language |
|
Point 1: For the micro-organisms, choose the correct abbreviation and annotate them in italic. |
|
Response 1: We thank the reviewer for pointing out this important detail. We have carefully reviewed the entire manuscript and corrected the formatting of all microbial names and their abbreviations to ensure they are presented in italics, in accordance with standard microbiological nomenclature. |
Round 2
Reviewer 3 Report
Comments and Suggestions for Authors
I am pleased to read that the authors addressed my comments and questions appropriately. I do not have any comments left.